# GRAPH-BASED VIRTUAL SENSING FROM SPARSE AND PARTIAL MULTIVARIATE OBSERVATIONS

**Giovanni De Felice**[1][*]**, Andrea Cini**[2]**, Daniele Zambon**[2]**, Vladimir V. Gusev**[1]**, Cesare Alippi**[2,3]
[1]University of Liverpool
[2]The Swiss AI Lab IDSIA & Università della Svizzera italiana
[3]Politecnico di Milano

## ABSTRACT

Virtual sensing techniques allow for inferring signals at new unmonitored locations by exploiting spatio-temporal measurements coming from physical sensors at different locations. However, as the sensor coverage becomes sparse due to costs or other constraints, physical proximity cannot be used to support interpolation. In this paper, we overcome this challenge by leveraging dependencies between the target variable and a set of correlated variables (covariates) that can frequently be associated with each location of interest. From this viewpoint, covariates provide partial observability, and the problem consists of inferring values for unobserved channels by exploiting observations at other locations to learn how such variables can correlate. We introduce a novel graph-based methodology to exploit such relationships and design a graph deep learning architecture, named GgNet, implementing the framework. The proposed approach relies on propagating information over a nested graph structure that is used to learn dependencies between variables as well as locations. GgNet is extensively evaluated under different virtual sensing scenarios, demonstrating higher reconstruction accuracy compared to the state-of-the-art.

## 1 INTRODUCTION

Spatio-temporal data analysis plays a significant role in applied and fundamental domains where systems evolve over both space and time. These include, for example, environmental science, urban planning, and epidemiology (Cressie & Wikle, 2015; Wang et al., 2020). In practice, the acquisition of spatio-temporal data is inevitably affected by all of the typical issues arising in real-world scenarios. Sensor and communication failures, for instance, can result in partial or even complete data loss at certain locations (Little & Rubin, 2019). Furthermore, the deployment of physical sensors can incur high costs that often limit the number of monitored locations. Nonetheless, inferring the values of observables at new target locations holds a significant value for analysis and, eventually, decision-making (Zhang et al., 2022; Hu et al., 2023). To address these challenges, *virtual sensing* (Liu et al., 2009) (also known as *spatio-temporal kriging*) (Stein, 1999) has emerged as a viable solution. Virtual sensing consists of reconstructing complete signals at unobserved locations by utilizing data gathered from monitored locations during the same time frame (Paepae et al., 2021; Brunello et al., 2021). As an example, virtual sensing techniques can be used to estimate solar irradiance at a particular location by exploiting neighboring sensors (Jayawardene & Venayagamoorthy, 2016), allowing, for instance, to plan for the installation of a new PV power plant. Most works rely on spatial correlations and on interpolating observations at neighboring locations (Appleby et al., 2020; Wu et al., 2021a;b; Zheng et al., 2023). However, as the sensor coverage becomes sparser due to costs and/or other (e.g., ecological) concerns, the number of neighbors from which to extract useful information becomes limited or even null. This makes the spatial position less informative and limits the effectiveness of existing approaches. One such prominent example is material weathering testing (De Felice et al., 2022), where such sparsity, together with the total lack of any historical data regarding the variable to infer at the target location, makes virtual sensing bound to fail. Despite being quite common in the real world, this scenario of sparse data collection is still mostly overlooked in the scientific literature.

---

[*]Correspondence to `g.de-felice@liverpool.ac.uk`.

In this work, we focus on the multivariate setting and show how leveraging correlations between variables can mitigate the challenges posed by sparse data scenarios. We consider sparsely distributed sensors, each of which is associated with a multivariate time series (MTS). MTS can be partly unobserved, i.e., completely lack some of the channels. In such settings, the problem becomes that of reconstructing missing channels (target variables) from the available ones (covariates) at the same location and from observations at related sensors (Fig. 1).

This scenario is typical of many real-world applications; in fact, variables correlated to the reconstruction target are often available at each location, e.g., satellite weather data can be used as covariates to infer photovoltaic energy production or material degradation. The approach developed in this paper follows from two observations. First, the mutual dependencies between the target variables and the covariates can be learned from the dynamics observed at locations where the target variable is being monitored. Such dependencies can then be leveraged at the target location, where covariates provide partial observability, to infer the missing variable. Second, the target variables can be reconstructed by integrating information from sensors that, although physically distant, can be considered close (similar) in a certain functional latent space. Such similarities can be learned from the data. To solve the task while taking advantage of the above considerations, we design

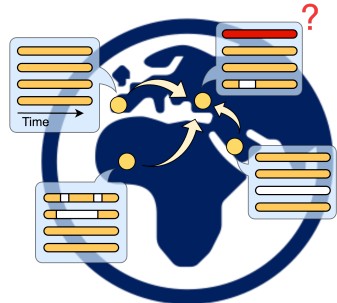

Figure 1: Sparse multivariate virtual sensing; available data (yellow), missing data (white), and predictions (red).

a novel graph-based framework allowing for modeling dependencies between variables and propagating information across locations through graph convolutional layers (Bronstein et al., 2021). Graph neural networks (GNNs) (Scarselli et al., 2008; Bacciu et al., 2020) indeed allow for exploiting dependencies as architectural biases in deep learning architectures and have recently found widespread success in time series processing (Li et al., 2018; Cini et al., 2023a; Jin et al., 2023). In particular, we propose a novel graph deep learning method, named Graph-graph Network (GgNet), designed to propagate information through a nested graph structure encompassing both sensors and channels. Relying on such a structure, the architecture of the resulting model is biased toward propagating representations by exploiting learned dependencies among both variables and locations. Summarizing, our novel contributions are as follows:

- we introduce a general methodology for performing multivariate virtual sensing, specifically designed to handle scenarios where available sensors do not provide adequate spatial coverage (Sec. 4.1);
- we propose a nested graph representation allowing for effectively modeling dependencies within multivariate spatio-temporal data (Sec. 4.2);
- we design GgNet, the first spatio-temporal graph neural network explicitly tailored for multivariate virtual sensing from sparse observations (Sec. 4.3);
- we carry out an extensive empirical evaluation by exploring different use cases and assessing the performance of the proposed method against the state-of-the-art (Sec. 5).

The proposed framework constitutes a first attempt at tackling a challenging setting for deep virtual sensing methods and arguably constitutes a key contribution to the methodological advancement of the field, as well as a powerful tool in practical applications.

## 2 PROBLEM FORMULATION

This section introduces the notation and formalizes the multivariate virtual sensing problem, with a focus on settings where sensors provide a sparse coverage of the area of interest.

**Notation** We indicate scalar variables and indices as lowercase $k$, constants as uppercase $K$, vectors as bold lowercase $\mathbf{x}$, matrices as bold uppercase $\mathbf{X}$, and higher order tensors and sets with calligraphic $\mathcal{X}$. We use the notation $\vec{\mathbf{x}}$ and $\vec{\mathbf{X}}$ to indicate univariate and multivariate time series, respectively. Association with the n-th location is indicated with the notation $x[n]$. Measurements at time step $t$ are indicated with subscript $x_t$; channel (variable) $d$ of a multivariate observation is indicated with superscript $x^d$.

**Multivariate spatio-temporal data** Spatio-temporal data refers to a collection of temporal observations coming from $N$ distinct spatial locations with coordinates $\{\mathbf{q}[n]\}_{n=1}^{N}$ in a generic domain $\Omega$, e.g., the one induced by the geographic placement of sensors. For each spatial location $n$, consider a multivariate time series (MTS) $\vec{\mathbf{X}}[n] \in \mathbb{R}^{T[n] \times D}$, where $T[n]$ and $D$ are, respectively, the number of time steps at location $n$, and the number of channels, i.e., observable variables. If observations are synchronous and an equal number of time steps are recorded across locations, the set of measurements can be represented as a 3-D tensor $\mathcal{X} \in \mathbb{R}^{N \times T \times D}$. Otherwise, the tensor representation can usually be obtained by padding or interpolating the missing observations (Lepot et al., 2017).

**Univariate virtual sensing** Consider a set of $N$ sensors positioned at distinct locations in a spatial domain $\Omega$ where a subset of $\bar{N} < N$ univariate time series $\{\vec{\mathbf{x}}[n] \in \mathbb{R}^{T}\}_{n=1}^{\bar{N}}$ are observed, while the remaining are missing. In line with previous works (Cao et al., 2018; Cini et al., 2022), we model data availability with a binary mask $m[n] \in \{0, 1\}$, which indicates if the $n$-th location is observed ($m[n] = 1$) or missing ($m[n] = 0$). The task of *univariate virtual sensing* consists of estimating $p(\vec{\mathbf{x}}[n] \,|\, \mathbf{q}[n], \mathcal{X})$, where time series $\vec{\mathbf{x}}[n]$ is unavailable (i.e., $m[n] = 0$), and the position $\mathbf{q}[n]$ of the associated sensor is given together with the *observation set* $\mathcal{X} = \{(\vec{\mathbf{x}}[n], \mathbf{q}[n]) \,|\, m[n] = 1\}_{n=1}^{N}$. Note that spatial coordinates are assumed to be available everywhere.

**Multivariate virtual sensing** The problem can easily be extended to the multivariate case. Here, consider a set of multivariate sensors $\{\vec{\mathbf{X}}[n] \in \mathbb{R}^{T \times D}\}_{n=1}^{N}$ where individual channels are missing at some locations. We can denote channel availability by extending the previous binary mask to $m^d[n] \in \{0, 1\}$, which indicates if the $d$-th channel is available at the $n$-th location. The *observation set* now consists of all the observed channels at all locations $\mathcal{X} = \{(\vec{\mathbf{x}}^d[n], \mathbf{q}[n]) \,|\, m^d[n] = 1\}_{n=1}^{N}$. The task of *multivariate virtual sensing* (MVS) then consists of modeling:

$$p(\vec{\mathbf{x}}^d[n] \,|\, \mathbf{q}[n], \mathcal{X}) \tag{1}$$

for every pair $(n, d)$ such that $m^d[n] = 0$. Note that virtual sensing is, a priori, a much more challenging task than the ordinary missing data imputation problem, due to the absence of any historical data to infer both the scale and the dynamics of the missing signal. When addressing this problem, the sensors' spatial proximity plays a pivotal role. If the sensors densely cover space, it is possible, in both the univariate and multivariate settings, to leverage the spatial proximity to available observations. However, the problem is far more challenging in sparse settings: if the problem is univariate, no observations are available at the target or any neighboring location. Conversely, the multivariate settings might allow for dealing with the virtual sensing problem by exploiting dependencies on variables other than the target one.

## 3 RELATED METHODS

Many approaches to kriging and virtual sensing have been proposed in geospatial analysis and machine learning literature (Cressie & Wikle, 2015). Notably, among existing methods, some rely on Gaussian processes (Luttinen & Ilin, 2012) and tensor decomposition (Bahadori et al., 2014). More related to our approach, Cini et al. (2022) and Marisca et al. (2022) introduced imputation methods based on GNNs and showed that such methods can perform virtual sensing as well. Wu et al. (2021a) and Zheng et al. (2023) directly tackle the virtual sensing problem exploiting inductive GNNs. Indeed, GNNs are becoming popular in time series imputation and reconstruction (Ye et al., 2021; Kuppannagari et al., 2021; Chen et al., 2022; Jin et al., 2023). However, all these approaches focus on univariate settings. As a consequence, they can only rely on the spatial proximity of the available sensors to perform the reconstruction and/or do not target sparser settings. Besides virtual sensing, deep learning methods have been widely applied in the context of MTS imputation by relying on a diverse range of model architectures, e.g., autoregressive RNNs (Cao et al., 2018), attention-based models (Shukla & Marlin, 2021; Du et al., 2023), generative adversarial networks (Yoon et al., 2018a; Luo et al., 2019) and diffusion models (Tashiro et al., 2021). Although such approaches can in principle be used to perform virtual sensing in the multivariate case, they cannot take spatial correlations directly into account. To the best of our knowledge, the proposed GgNet is the first graph-based neural network architecture designed to tackle MVS in sparse scenarios by exploiting learned relationships across both observable variables and locations. We refer to Appendix A for a more thorough discussion of the relevant literature for spatio-temporal kriging and MTS imputation.

## 4 METHODOLOGY

Here, we introduce a general framework for solving the MVS problems formalized in Sec. 2, regardless of the spatial coverage offered by the available sensors. We begin by presenting a conceptualization of the inference problem as dealt with in our framework. Then, we introduce the details of the proposed novel architecture, aligned with such conceptualization.

### 4.1 CONCEPTUALIZATION

We identify two fundamental types of dependencies characterizing the MVS task, i.e., the relationships between channels at the same location (*intra* location) and among observations at different locations (*inter* location). We model such dependencies by considering location-channel $(n, d)$ pairs and defining the sets

$$\mathcal{Y}^d[n] = \left\{ \vec{\mathbf{x}}^\delta[\nu] \mid \delta = d \,\wedge\, \nu = n \,\wedge\, m^\delta[\nu] = 0 \right\} \qquad \text{(target set)}$$

$$\mathcal{T}^d[\backslash n] = \left\{ \vec{\mathbf{x}}^\delta[\nu] \mid \delta = d \,\wedge\, \nu \neq n \,\wedge\, m^\delta[\nu] = 1 \right\} \qquad \text{(observed target set)}$$

$$\mathcal{C}^{\backslash d}[n] = \left\{ \vec{\mathbf{x}}^\delta[\nu] \mid \delta \neq d \,\wedge\, \nu = n \,\wedge\, m^\delta[\nu] = 1 \right\} \qquad \text{(intra-location covariate set)}$$

$$\mathcal{C}^{\backslash d}[\backslash n] = \left\{ \vec{\mathbf{x}}^\delta[\nu] \mid \delta \neq d \,\wedge\, \nu \neq n \,\wedge\, m^\delta[\nu] = 1 \right\} \qquad \text{(inter-location covariate set)}$$

where $\backslash n$ indicates all indices except $n$. Sets $\mathcal{Y}^d[n], \mathcal{T}^d[\backslash n], \mathcal{C}^{\backslash d}[n]$ and $\mathcal{C}^{\backslash d}[\backslash n]$ encompass, respectively, the target variable at the target location, the target variables at all other locations except $n$, the observed variables at the target location, and the observed variables at all other locations except $n$. By considering the above sets, we distinguish between the types of dependencies that allow for reconstructing a missing channel $\mathbf{x}^d[n]$ in the *target set*.

**T→Y:** Relations between observed targets at other locations ($\mathcal{T}^d[\backslash n]$) and the target $\mathbf{x}^d[n]$.

**C→Y:** Relations between observed covariates at the target location ($\mathcal{C}^{\backslash d}[n]$) and the target $\mathbf{x}^d[n]$.

The subtask of learning from T→Y dependencies is similar to learning how to exploit observations at related locations typical of univariate virtual sensing. To weigh the importance of different observations in the *observed target set* w.r.t. the target, a notion of similarity between locations is required. Kriging methods address it by relying on spatial proximity (dense scenario), e.g., as derived from the sensor positions $\{\mathbf{q}[n]\}$, to weigh the reciprocal importance of observations at different locations. In contrast, with the sparse settings in mind, we model T→Y dependencies in a data-driven fashion, learning a similarity score for each pair of locations. In particular, as detailed in Sec. 4.2, we associate a representation (node embedding) (Cini et al., 2023a) to each location and learn a score function taking as input pairs of such representations. At the same time, these are used as additional learned covariates to tailor C→Y in modeling location-specific dynamics. Such a framework allows for the extension of virtual sensing to generic MTS datasets, as it does not rely on pre-defined spatial relationships such as physical proximity. In the following, we design a model that directly accounts for both types of dependencies with architectural inductive biases that align the processing with the modeling of intra and inter-location dependencies.

### 4.2 NESTED GRAPH STRUCTURE

To take advantage of both T→Y and C→Y relations, we learn a nested graph structure composed of an inter-location graph and an intra-location graph (Fig. 2). The inter-location graph connects different spatial locations, while the intra-location one explicits dependencies among channels.

**Inter-location graph $G$:** We associate each sensor location to a node of a graph $G$, which we call *inter-location* graph, and model the association between different sensors as edges of $G$, thus accounting for the T→Y modeling.

As physical proximity cannot be exploited in sparse settings, we learn the graph topology ($N \times N$ adjacency matrix $\mathbf{A}_G$) from the data to account for dependencies among sensors that can be far apart in space. In particular, following previous works on local representations (Cini et al., 2023a), we

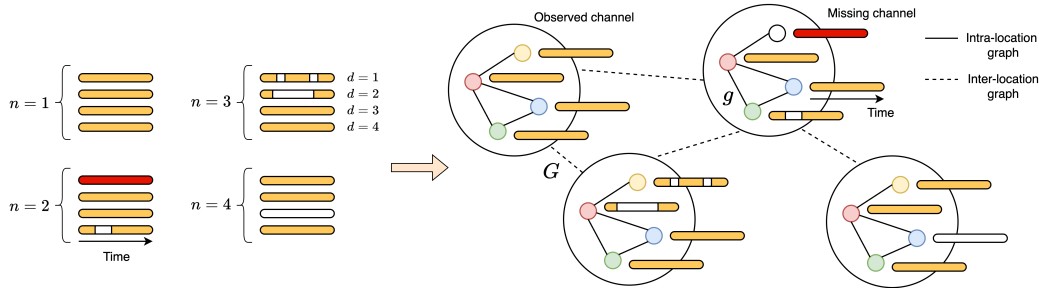

Figure 2: Dataset as a collection of multivariate time series (left) and its nested graph representation (right). Each location is represented as a node in the inter-location graph (G), capturing generic relations between locations. Within each node in G, a smaller intra-location graph (g) models dependencies between the channels.

associate each $n$-th node in $G$ with a learnable (static) embedding $\mathbf{e}_G[n] \in \mathbb{R}^{h_{e,G}}$. These are learned as parameters, jointly with the network weights, and are used to estimate relationships relevant to the downstream MVS task. The weighted adjacency matrix is then obtained from the similarities between embeddings. In particular, we model $\mathbf{A}_G$ as

$$\mathbf{A}_G = Softmax(\text{MLP}_1(\mathbf{E}_G) \cdot \text{MLP}_2(\mathbf{E}_G)^T) \tag{2}$$

where the matrix $\mathbf{E}_G \in \mathbb{R}^{N \times h_{e,G}}$ contains the learnable node embeddings for all nodes in $G$. Convolutions on such a graph directly account for T→Y dependencies by propagating information across locations. Similarly, we learn an adjacency matrix for intra-location dependencies.

**Intra-location graph $g$:** We identify each of the $D$ channels, at every $n$-th sensor, with a node of a second graph, which we name *intra-location* graph $g$, accounting for the relationships between channels (C→Y).

Similarly to inter-location case, we associate each $d$-th node in $g$ with a learnable embedding $\mathbf{e}_g[d]$. We learn adjacency matrix $\mathbf{A}_g \in \mathbb{R}^{D \times D}$ of $g$ as a function of a $D \times D$ matrix $\Phi$ of free parameters, treated as edge scores. As the physical quantities considered at each location are the same, we assume $g$ to be the same in all locations.

Note that, in this paper, for simplicity, we do not consider graphs $G$ and $g$ that change over time and do not constrain $G$ and $g$ to be discrete binary objects. That being said, more advanced graph learning methods (e.g., Kipf et al. 2018; Niculae et al. 2023; Cini et al. 2023b) can be incorporated into the framework and constitute a possible extension for future works. Note that the node embeddings $\mathbf{e}_G[n]$ need to be learned for every location, which makes the model intrinsically *transductive*. Despite that, the fine-tuning of new node embeddings can be done efficiently (Cini et al., 2023a).

### 4.3 GRAPH-GRAPH NETWORK

In this section, we present the *Graph-graph Network* (GgNet), our proposed architecture relying on the nested graph representations provided by $G$ and $g$. The model is composed of several blocks, as detailed below and depicted in Fig. 3.

**Input encoder** To account for specifics along the temporal, spatial and channel dimensions, we obtain a hidden representation for each time step $t$, location $n$, and channel $d$ given the input data $\mathcal{X}$ and the learned node embeddings $\mathbf{E}_G$, $\mathbf{E}_g$ similarly to Marisca et al. (2022)

$$\mathbf{h}_t^d[n] = m_t^d[n] \cdot \text{MLP}_{\text{enc},1}(\mathbf{x}_t^d[n]) + \text{MLP}_{\text{enc},2}(\mathbf{e}_G[n], \mathbf{e}_g[d]) \tag{3}$$

where $\mathbf{h}_t^d[n] \in \mathbb{R}^H$ are the resulting encoded representations, while $\text{MLP}_{\text{enc},1}$ and $\text{MLP}_{\text{enc},2}$ are two distinct multilayer perceptions. Note that representations are obtained by exploiting observations wherever available and relying exclusively on node embeddings for the targets.

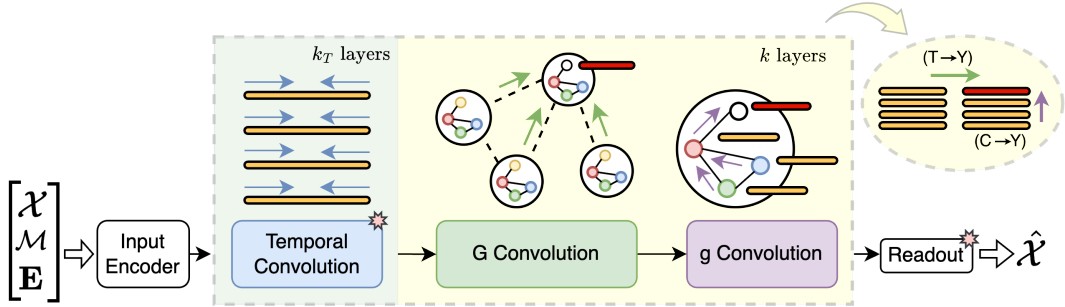

Figure 3: Overview of GgNet. Temporal convolutions (blue) encode temporal patterns, $G$-convolutions (green) propagate information across the inter location graph and model dependencies between locations; $g$-convolutions (purple) propagate information across the intra-location graph and model dependencies between channels. Starred modules refer to channel-wise operations.

**Temporal processing**  Stacks of Temporal Convolutions (TC) (Bai et al., 2018) are used to encode temporal information for each location and channel as

$$\vec{\mathbf{H}}^d[n] = \mathrm{TC}^d\left(\vec{\mathbf{H}}^d[n], \vec{\mathbf{m}}^d[n], \mathbf{e}_G[n]\right) \tag{4}$$

where $\vec{\mathbf{H}}^d[n] \in \mathbb{R}^{T \times H}$. As channels may present largely heterogeneous temporal features, we use a different set of weights (filters) for processing data in each channel. Temporal convolutions are highly parallelizable making them preferable, in this setting, over, e.g., recurrent neural networks. Furthermore, we symmetrically pad the input sequence to avoid duplicating the architecture to encode forward and backward dynamics and exploit exponentially increasing dilation rates to capture both short and long-range temporal dependencies.

**Inter-location (spatial) processing**  Spatial information is propagated across locations (T→Y) by means of Graph Convolutions (GC) (Bronstein et al., 2021) over the inter-location graph ($G$-*convolution* in Fig. 3):

$$\mathbf{H}_t^d = \mathrm{GC}_{(G)}\left(\mathbf{H}_t^d, \mathbf{m}_t^d, \mathbf{E}_G, \mathbf{E}_g; \mathbf{A}_G\right) \tag{5}$$

where $\mathbf{H}_t^d \in \mathbb{R}^{N \times H}$ and the same convolution is performed independently on each channel and each time step. Specifics of each channel are accounted for via the conditioning on the embeddings in $\mathbf{E}_g$. Note that *G-convolution* is a synchronous operation that assumes that the time frames of each time series are aligned.

**Intra-location (channel-wise) processing**  As the next step, information is propagated across channels (C→Y) by performing GCs over the intra-location graph ($g$-*convolution* in Fig. 3):

$$\mathbf{H}_t[n] = \mathrm{GC}_{(g)}\left(\mathbf{H}_t[n], \mathbf{m}_t[n], \mathbf{E}_G, \mathbf{E}_g; \mathbf{A}_g\right) \tag{6}$$

for all time steps and locations, and where $\mathbf{H}_t[n] \in \mathbb{R}^{D \times H}$. *g-convolutions* allow for inferring observations at unavailable channels by modeling dependencies among targets and covariates. Note that weights are shared across locations, as local dynamics are accounted for by conditioning on node embeddings $\mathbf{E}_G$.

Multiple $T$-, $G$-, and $g$-*convolution* blocks can be stacked, allowing for designing deep architectures. We refer to Appendix E.2 for an extensive ablation study assessing the impact of each component.

**Readout**  Finally, a readout composed of $d$ MLPs maps representations to predictions at all target locations and time steps as

$$\hat{\mathbf{x}}_t^d[n] = \mathrm{MLP}_{\mathrm{dec}}^d(\mathbf{h}_t^d[n], m_t^d[n], \mathbf{e}_G[n], \mathbf{e}_g[d]). \tag{7}$$

Note that the GgNet implementation considered here is just one of many possible instantiations of the framework. The model can be extended, e.g., by including more sophisticated graph processing and learning modules.

### 4.4 MODEL TRAINING

For a generic MVS task, the reconstruction error can be defined as

$$\mathcal{L}(\hat{\mathcal{X}}, \mathcal{X}, \mathcal{M}) = \frac{\sum_n \sum_d \bar{m}^d[n] \cdot \ell(\hat{\vec{\mathbf{x}}}^d[n], \vec{\mathbf{x}}^d[n])}{\sum_n \sum_d \bar{m}^d[n]} \qquad (8)$$

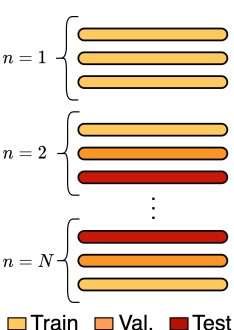

where $\hat{\vec{\mathbf{x}}}^d[n]$ and $\vec{\mathbf{x}}^d[n]$ are, respectively, the predicted and true values, and $\ell(\cdot, \cdot)$ is a loss function between time series, e.g., the mean absolute error or the mean squared error. The model is trained and evaluated by simulating the presence of missing channels partitioning the available $N \times D$ channels into training, validation, and test channels (Fig. 4), simulating a *transductive* learning setting. To bias the training toward the virtual sensing task, we mask out a fraction of the available channels for each training batch and train the model to reconstruct the input data, giving a higher weight to the masked out observations. We have found this strategy to be crucial for models like GgNet that do not rely on forecasting as a surrogate objective.

Figure 4: Training and evaluation splits for MVS.

Finally, we mask out an additional small fraction of data points, at random, in the training data to enforce robustness to random missing values (Cini et al., 2022; Du et al., 2023).

## 5 EXPERIMENTAL EVALUATION

In this section, we evaluate the proposed method in several MVS tasks. Experiments are conducted across three different datasets, allowing for the investigation of settings characterized by (i) different temporal resolutions and (ii) different degrees of sparsity, as well as (iii) assessing the limitations of the different approaches. Alongside GgNet, we provide results for the following baselines: **KNN**, i.e., a simple baseline averaging the nearest geographical neighbors, **BRITS** (Cao et al., 2018) as representative of autoregressive RNN global models[1], **SAITS** (Du et al., 2023) as representative of transformer-based global models unaware of spatial relations, **GRIN** (Cini et al., 2022) as representative of graph-based imputation models, leveraging spatial correlations. Moreover, we consider a variation of the standard GRIN model, named **GRIN**$_m$, modified to better deal with multivariate data at each location; specifically, we add self-loops in GRIN's *Spatial Decoder* for conditioning on covariates at the same time step of the reconstruction target. To assess the impact of some of the introduced design choices, we also compare 4 progressively more advanced recurrent architectures: 1) **RNN**, a standard global recurrent neural network; 2) **RNN**$_{\text{BiD.}}$, its bidirectional extension; 3) **RNN**$_{\text{Emb.}}$, which, besides being bidirectional, incorporates a local component by concatenating learnable node embeddings to the input observations; 4) **RNN**$_{\text{G}}$, which adds a convolution operation over an inter-location graph on top of the previous architecture. For each model, we consider different hyperparameter configurations, with increasing complexity; reported results correspond to the best-performing model, given the reconstruction accuracy on the validation set. Regarding the preprocessing steps, we follow the common practice of standardizing the data, ensuring that, across locations and time, each of the $D$ channels has zero mean and unit variance. Missing values, if present, are masked out. Details about the baselines can be found in Appendix B, while the complete experimental setup is described in Appendix C.

### 5.1 MISSING-CHANNEL RECONSTRUCTION: CLIMATIC DATA

In the first experiment, we address reconstructing missing-at-random (MaR) channels in multivariate spatio-temporal datasets. We build two datasets with different temporal resolutions of climatic variables from the NASA Langley Research Center POWER Project. More information about the data and APIs is available in Appendix D. To emulate a reduced spatial coverage, we collect data in correspondence with the 235 national capitals. For each location, we collect multiple correlated variables with daily and hourly resolutions. Within this setting, we aim to reconstruct MaR climatic variables at different locations from the available data. Among all $N \times D$ channels, we use $70\%$ for training, $10\%$ for validation, and the remaining $20\%$ for testing.

---

[1]With the term "global" we refer to models whose parameters are shared in all locations, in contrast to "local" ones that have location-specific parameters (Montero-Manso & Hyndman, 2021; Cini et al., 2023a).

Table 1: Climatic dataset: channel-wise MRE (%) for daily data and average accuracy for daily (D) and hourly (H) data. Results are averaged across locations and 5 random seeds. The best-performing method is in bold, the second-best is underlined.

| CH. | TEMP. MEAN | TEMP. RANGE | TEMP. MAX | WIND SPEED | REL. HUM. | PREC. | TEMP. DEW | CLOUDS | IRR. SHORT | IRR. LONG | AVG (D) | AVG (H) |
|---|---|---|---|---|---|---|---|---|---|---|---|---|
| KNN | $15.6_{\pm2.5}$ | $42.1_{\pm6.3}$ | $12.4_{\pm1.5}$ | $33.7_{\pm4.5}$ | $12.3_{\pm0.4}$ | $102.8_{\pm4.7}$ | $23.9_{\pm2.9}$ | $34.0_{\pm0.6}$ | $18.0_{\pm0.5}$ | $6.0_{\pm1.0}$ | $30.1_{\pm1.0}$ | $36.2_{\pm1.9}$ |
| BRITS | $4.0_{\pm0.6}$ | $17.5_{\pm2.3}$ | $3.3_{\pm0.5}$ | $32.1_{\pm1.5}$ | $5.5_{\pm0.8}$ | $80.5_{\pm2.4}$ | $8.4_{\pm1.1}$ | $29.9_{\pm0.9}$ | $20.9_{\pm1.0}$ | $3.4_{\pm0.4}$ | $20.5_{\pm0.5}$ | $26.3_{\pm0.8}$ |
| GRIN | $4.2_{\pm0.7}$ | $18.6_{\pm1.9}$ | $4.3_{\pm0.5}$ | $30.7_{\pm2.3}$ | $5.6_{\pm0.6}$ | $77.3_{\pm3.8}$ | $8.7_{\pm1.0}$ | $29.1_{\pm0.3}$ | $15.3_{\pm0.7}$ | $3.9_{\pm0.3}$ | $19.8_{\pm0.6}$ | $23.3_{\pm0.7}$ |
| GRIN$_m$ | $2.9_{\pm0.9}$ | $12.9_{\pm1.3}$ | $2.8_{\pm0.6}$ | $30.7_{\pm3.9}$ | $3.6_{\pm0.9}$ | $69.9_{\pm1.6}$ | $6.4_{\pm1.1}$ | $\underline{20.5}_{\pm0.9}$ | $\underline{11.7}_{\pm0.6}$ | $3.6_{\pm0.4}$ | $16.5_{\pm0.6}$ | $22.7_{\pm0.4}$ |
| SAITS | $\underline{2.4}_{\pm0.4}$ | $\underline{11.2}_{\pm1.3}$ | $\underline{2.3}_{\pm0.4}$ | $\underline{26.9}_{\pm1.0}$ | $\underline{3.3}_{\pm1.0}$ | $\underline{66.0}_{\pm2.5}$ | $\underline{5.0}_{\pm1.0}$ | $20.6_{\pm0.7}$ | $14.2_{\pm0.9}$ | $\mathbf{2.8}_{\pm0.3}$ | $\underline{15.5}_{\pm0.5}$ | $\underline{22.2}_{\pm0.5}$ |
| GGNET | $\mathbf{2.1}_{\pm0.4}$ | $\mathbf{9.6}_{\pm0.7}$ | $\mathbf{2.0}_{\pm0.2}$ | $\mathbf{23.9}_{\pm2.2}$ | $\mathbf{2.7}_{\pm0.7}$ | $\mathbf{60.6}_{\pm2.1}$ | $\mathbf{4.2}_{\pm0.8}$ | $\mathbf{16.5}_{\pm0.5}$ | $\mathbf{9.2}_{\pm0.8}$ | $\underline{2.9}_{\pm0.3}$ | $\mathbf{13.4}_{\pm0.2}$ | $\mathbf{20.4}_{\pm0.6}$ |
| % IMP. | 12.5% | 14.3% | 13.0% | 11.1% | 18.1% | 8.1% | 16.0% | 19.9% | 21.4% | -3.6% | 13.5% | 8.1% |
| RNN | $5.6_{\pm0.7}$ | $21.9_{\pm0.7}$ | $5.8_{\pm0.5}$ | $32.7_{\pm1.2}$ | $6.9_{\pm0.7}$ | $83.8_{\pm1.8}$ | $10.8_{\pm1.1}$ | $35.5_{\pm1.0}$ | $21.8_{\pm0.3}$ | $3.8_{\pm0.4}$ | $22.9_{\pm0.2}$ | $28.8_{\pm0.8}$ |
| + BID. | $4.2_{\pm0.6}$ | $19.1_{\pm1.0}$ | $4.7_{\pm0.4}$ | $31.1_{\pm0.8}$ | $5.9_{\pm0.6}$ | $78.0_{\pm2.6}$ | $8.6_{\pm1.0}$ | $31.4_{\pm0.6}$ | $19.0_{\pm0.4}$ | $3.6_{\pm0.4}$ | $20.6_{\pm0.3}$ | $25.7_{\pm0.5}$ |
| + EMB. | $4.1_{\pm0.5}$ | $17.5_{\pm1.5}$ | $4.3_{\pm0.4}$ | $30.2_{\pm2.0}$ | $5.5_{\pm0.6}$ | $77.1_{\pm1.6}$ | $8.5_{\pm1.0}$ | $30.2_{\pm0.4}$ | $17.5_{\pm0.7}$ | $3.6_{\pm0.3}$ | $19.8_{\pm0.5}$ | $26.2_{\pm0.9}$ |
| + G | $3.6_{\pm0.3}$ | $16.5_{\pm1.3}$ | $3.9_{\pm0.4}$ | $27.6_{\pm1.6}$ | $5.2_{\pm0.4}$ | $74.7_{\pm1.8}$ | $7.3_{\pm0.9}$ | $27.7_{\pm0.5}$ | $14.0_{\pm0.8}$ | $3.6_{\pm0.4}$ | $18.4_{\pm0.5}$ | $23.2_{\pm1.1}$ |

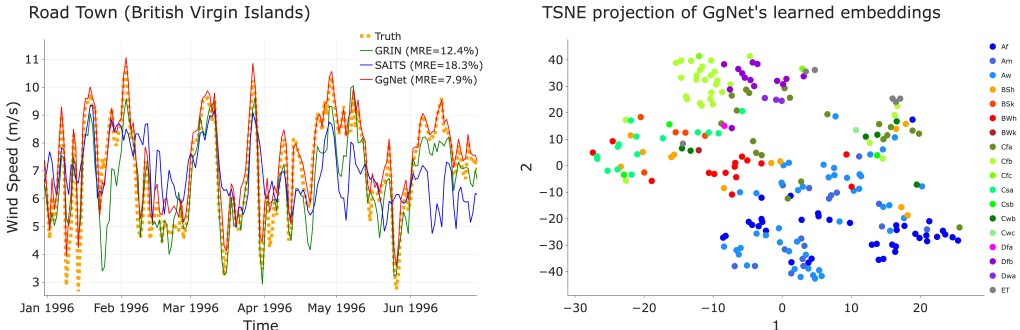

Figure 5: (Left) Reconstruction of *wind speed* (poorly correlated with the other channels) in Road Town (which can exploit observations at nearby Caribbean capitals). (Right) t-SNE representation of the node embeddings learned by GgNet; colours refer to the Köppen-Geiger climate classification.

**Daily climatic data** Considering a daily resolution, we collect 10 climatic variables (listed in Appendix D) encompassing 30 years, which results in $N = 235, T = 10958, D = 10$. Tab. 1 reports the Mean Relative Error (MRE) for each test channel, grouped by variable and averaged across locations. All results are consistent across different metrics as shown in Appendix E.8. From the results, we first observe that GgNet outperforms models relying exclusively on pre-defined spatial relations (GRIN), demonstrating superior performance in handling sparse settings. Notably, GRIN underperforms especially in correspondence with remote locations, where no neighboring sensor is available (Appendix E.3). The GRIN$_m$ variant improves over the results of GRIN, but nonetheless underperforms w.r.t. GgNet. SAITS performs reasonably well on this dataset, ranking second in most of the setups. In particular, SAITS can account for intra-location covariates, but cannot take advantage of inter-location relations. This is particularly evident for missing channels that are hard to infer from the other variables at the same location (e.g., *wind speed*, *precipitations*, *clouds* and *shortwave irradiation*), but can be reconstructed from spatially close observations, as in the example shown in Fig. 5 (left). Tab. 1's lower section reports results for the RNN model variants. The notable performance improvement obtained by the bidirectional variant demonstrates the importance of accounting for both past and future values. Node embeddings yield modest yet consistent improvements. Additionally, the introduction of the inter-location graph G results in a substantial performance gain, supporting the importance of inter-location relations. Alongside MVS results, we provide in Fig. 5 (right) a t-SNE visualization of the node embeddings ($\mathbf{E}_G$, Sec. 4.2) as learned by GgNet w.r.t. each location. Each embedding is coloured according to the Köppen-Geiger climate classification (Peel et al., 2007) that divides the globe into different climates, with similar colours corresponding to similar climates. Note that the model does not have access, at any point, to either these climate labels or to the geographical coordinates of any location. The correspondence between

Table 2: Photovoltaic dataset: MAE (W*h). For each N, results are averaged across 5 sampling of the locations and 5 different runs for each set (25 runs in total). The best-performing method is in bold, the second-best is underlined.

| $N_{LOCATIONS}$ | 10 | 20 | 30 | 50 | 70 | 100 | 150 | 200 |
|---|---|---|---|---|---|---|---|---|
| GRIN | $581 \pm 170$ | $536 \pm 52$ | $533 \pm 49$ | $514 \pm 36$ | $531 \pm 34$ | $524 \pm 28$ | $523 \pm 40$ | $528 \pm 43$ |
| $GRIN_m$ | $497 \pm 117$ | $458 \pm 49$ | $\underline{446} \pm 42$ | $\underline{424} \pm 23$ | $\underline{428} \pm 21$ | $423 \pm 13$ | $\underline{417} \pm 13$ | $418 \pm 12$ |
| SAITS | $\underline{466} \pm 68$ | $\mathbf{451} \pm 46$ | $448 \pm 37$ | $427 \pm 29$ | $436 \pm 24$ | $\underline{420} \pm 16$ | $\underline{417} \pm 14$ | $\underline{416} \pm 14$ |
| **GGNET** | $\mathbf{452} \pm 67$ | $\underline{455} \pm 39$ | $\mathbf{439} \pm 32$ | $\mathbf{415} \pm 26$ | $\mathbf{410} \pm 26$ | $\mathbf{391} \pm 18$ | $\mathbf{367} \pm 15$ | $\mathbf{357} \pm 16$ |
| % IMP. | 3.0% | -0.9% | 1.6% | 2.1% | 4.2% | 6.9% | 12.0% | 14.2% |

such labels and the formed clusters serves as a qualitative assessment of the capability of GgNet to capture similarities in the observed dynamics reflecting the climate at each location. Appendix E reports several additional analyses.

**Hourly climatic data**  Then, we address the same MaR task in the case of 7 climatic variables sampled with hourly resolution. Data cover one year, which results in $N = 235, T = 8760, D = 7$. Differently from daily data, hourly data are more difficult to predict, being characterized by high-frequency components both temporally and spatially. As a consequence, exploiting inter-location dependencies can help in providing better conditioning and hence lead to more accurate predictions. Average MRE is reported in the last column of Tab. 1, while channel-wise results are shown in Appendix E.8. Graph-based models effectively exploit the additional information to obtain a relatively good reconstruction accuracy. In GgNet, accounting for the interplay between spatial aggregation (T→Y) and global modeling of the relations across channels (C→Y) arguably results in superior average performance.

## 5.2 EXTENDING SPATIAL COVERAGE: PHOTOVOLTAIC ENERGY PRODUCTION

We consider the problem of predicting a specific target variable (whose direct observation is expensive) at new target locations by exploiting a set of available covariates. This constitutes an interesting use case for our multivariate virtual sensing task. For example, renewable energy production or material degradation can be inferred from widely available satellite climatic data (De Felice et al., 2022). In particular, we consider as target variable the daily photovoltaic power (PV) production simulated over continental North America by Hu et al. (2022). To simulate a sparse scenario, we randomly sample a few locations from the provided dense grid. As covariates, we use the same 10 climatic variables from Sec. 5.1 collected in correspondence with the sampled locations. For both target and covariates, we consider 1 full year of daily observations, which results in $T = 365, D = 11$. By varying the number of locations between 10 and 200, we aim to infer the PV production at new target locations from the climate at such locations. We use 70% of the sensors for training, 10% for validation, and 20% for testing. Tab. 2 reports the reconstruction accuracy in terms of the Mean Absolute Error (MAE). As more locations are considered, results show a progressive increment of the relative improvement of GgNet over SAITS. This could be related to the superior capability of GgNet to exploit spatial information.

## 6 CONCLUSION

We presented a novel framework for virtual sensing from sparse multivariate spatio-temporal observations. In doing so, we introduced a novel methodology to exploit dependencies between covariates and relations across locations. In this context, we proposed GgNet, a graph deep learning architecture leveraging a nested graph structure to account for such relations. The relational structure is learned end-to-end to maximize reconstruction accuracy. Compared to state-of-the-art, GgNet can achieve superior performance in settings with poor sensor coverage, where other methods fail. As future directions, we believe it would be interesting to explore the application of a similar methodology to collections of asynchronous time series, typically found in healthcare applications. Finally, GgNet is limited, in its current form, to transductive learning settings and its extension to inductive learning would broaden the applicability of the proposed method.

REPRODUCIBILITY STATEMENT

Python code to reproduce the experiments is available online at `https://github.com/gdefe/ggnet-virtual-sensing`. A script to download climatic data from the database in correspondence with the world capitals is included in the provided code. PV datasets at different spatial densities, together with the corresponding climatic variables, are provided alongside the code. Complete climatic and photovoltaic datasets are publicly available online through the references provided in Appendix D. The sampling of locations (photovoltaic experiment) and simulation of missing channels (both climatic and photovoltaic experiment) are controlled by a fixed seed for reproducibility purposes. Hyperparameters for all models are provided in Appendix C.

ACKNOWLEDGMENTS

G.D.F. acknowledges the Beckers Group for funding this research. V.V.G. thanks Leverhulme Trust for support via the Leverhulme Research Centre for Functional Materials Design and EPSRC under grant number EP/V026887. This research was partly funded by the Swiss National Science Foundation under grant 204061: *High-Order Relations and Dynamics in Graph Neural Networks*. Climate data was obtained from the National Aeronautics and Space Administration (NASA) Langley Research Center (LaRC) Prediction of Worldwide Energy Resource (POWER) Project funded through the NASA Earth Science/Applied Science Program. The authors wish to thank the anonymous reviewers whose comments considerably improved the final version of the paper.

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

APPENDIX

## A  RELATED METHODS

Methods that can perform virtual sensing can be broadly categorized into spatio-temporal kriging methods and generic MTS imputation approaches. Kriging methods exploit the density of the network and infer the missing sensors by interpolating neighboring observations. Among those, ordinary kriging (Stein, 1999) is a linear interpolation procedure that assigns optimal weights for neighboring data points based on a variogram model. Similarly, cokriging (Goovaerts, 1998; Cressie & Wikle, 2015) also exploits correlation with auxiliary covariate information (such as elevation, soil type, or temperature) and estimates them simultaneously with the variable of interest. However, these are linear spatial interpolation methods, therefore, they fail to capture any temporal patterns, as well as nonlinear dependencies within the data. Multi-task Gaussian Processes leverage flexible kernel structures to capture spatio-temporal correlations (Bonilla et al., 2007; Luttinen & Ilin, 2012). In addressing scalability issues when dealing with extensive datasets, matrix/tensor completion methods capture global consistency in the data by imposing a low-rank assumption, as well as local consistency by diverse key regularization structures (Bahadori et al., 2014; Takeuchi et al., 2017; Lei et al., 2022; Wu et al., 2022). More related to our approach, imputation methods based on GNNs have demonstrated great effectiveness in modeling spatial dependencies between time series (Ye et al., 2021; Kuppannagari et al., 2021; Roth & Liebig, 2022; Kong et al., 2023; Jin et al., 2023). Notably, GRIN (Cini et al., 2022) accounts for both temporal and spatial correlation within the data by combining autoregressive modeling with message-passing operations; SPIN (Marisca et al., 2022) introduces a spatiotemporal attention mechanism to weigh discrete points in time and space; IGNNK (Wu et al., 2021a) tackle inductive virtual sensing by specializing spatio-temporal graph neural networks, later extended in SATCN (Wu et al., 2021b) by allowing multiple transformations of the spatial information to contribute to the adjacency matrix; INCREASE (Zheng et al., 2023) also considers different heterogeneous spatial relations. However, they all assume that the spatial information characterizing the placement of each sample is available to the model; either as a given graph structure or as geographical information to compute one. In sparse settings, or when such information is missing, the main limitation of these methods is deriving an appropriate adjacency matrix. This problem is even exacerbated in the multivariate case. As evidence, MTS similarity is a challenging problem on its own (Mikalsen et al., 2018; De Felice et al., 2023). Adaptive methods (Bai et al., 2020), such as AGRN (Chen et al., 2022), address this by learning the adjacency matrix together with the network weights. Despite this, existing GNN-based approaches only target dense and univariate sensor networks and do not explicitly model the relations between channels in individual locations.

On the other hand, recent advancements in generic MTS imputation have led to models that can be adapted to perform virtual sensing. Among these, the most widely adopted are deep autoregressive methods based on recurrent neural networks (Che et al., 2018; Yoon et al., 2018b), e.g. BRITS (Cao et al., 2018) is a bidirectional model that also takes advantage of the relationships between dimensions. Moreover, imputation can be performed by modeling the underlying data distribution, generally employing generative adversarial neural networks (Yoon et al., 2018a; Luo et al., 2019). In recent times, imputation techniques have been proposed that rely on attention mechanisms (Ma et al., 2019; Shukla & Marlin, 2021; Du et al., 2023) and diffusion models (Tashiro et al., 2021). However, these models cannot take spatial correlations directly into account. Indeed, these methods learn a single model shared across locations and take into account a single MTS at a time. One could still exploit such approaches by considering all the available data as a single MTS, but the resulting model would not be appropriate for virtual sensing, as the same variables at different locations would be modeled as entirely different channels.

## B  BASELINE DETAILS

**KNN:**  imputes missing test channels by averaging the corresponding train channels in the $k$-nearest geographical neighbors. The number of neighbors $k \in \{1, 2, 3, 5, 10\}$ is selected based on the best performance on the validation set. This baseline has access to the geographical coordinates, with which neighbors are identified.

**RNN:** global model where data at different locations $\mathbf{X}[n] \in \mathbb{R}^{T \times D}$ are considered as different training instances to train a shared model. Specifically, this model imputes missing values from a linear transformation of the hidden state of a global Recurrent Neural Network (RNN). Formally, the following operations are sequentially iterated for every time step $t$ in the considered temporal window:

$$\hat{\mathbf{x}}_t = \mathbf{W}_{out} \, \mathbf{h}_t + \mathbf{b} \tag{9}$$

$$\mathbf{x}_t = \mathbf{m}_t \odot \mathbf{x}_t + \bar{\mathbf{m}}_t \odot \hat{\mathbf{x}}_t \tag{10}$$

$$\mathbf{h}_t = \text{GRU}( \, [\, \mathbf{x}_t \, || \, \mathbf{m}_t \,], \, \mathbf{h}_{t-1} \,) \tag{11}$$

where the notation $[n]$ is implicit for all inputs, masks and hidden states.

**RNN$_{\text{BiD}}$:** this model replicates the previous RNN model both in the forward direction and in the backward direction. Imputations performed by Eq. 9 are discarded and the final imputation is obtained with a separate MLP acting on the aggregated hidden representations:

$$\hat{\mathbf{x}}_t = \text{MLP}( \, [\, \mathbf{h}_t^{fwd} \, || \, \mathbf{h}_t^{bwd} \,] \,) \tag{12}$$

**RNN$_{\text{emb}}$:** as global models, it is not possible for the two previous models to specialize the modeling on the specifics of each location. In this extension, we incorporate local components into RNN$_{\text{BiD}}$ by concatenating the node embeddings $\mathbf{e}[n]$ to all input time steps before the state update and readout. Formally, the model sequentially iterates, on both directions, for every time step $t$, Eq. 9, Eq. 10 and:

$$\mathbf{h}_t[n] = \text{GRU}( \, [\, \mathbf{x}_t[n] \, || \, \mathbf{m}_t[n] \, || \, \mathbf{e}[n] \,], \, \mathbf{h}_{t-1}[n] \,) \tag{13}$$

where we have here explicit the notation $[n]$. Finally, as before, we aggregate hidden representations to output the final result:

$$\hat{\mathbf{x}}_t[n] = \text{MLP}( \, [\, \mathbf{h}_t^{fwd}[n] \, || \, \mathbf{h}_t^{bwd}[n] \, || \, \mathbf{e}[n] \,] \,) \tag{14}$$

**RNN$_{\text{G}}$:** we further build here on the previous model by introducing synchronous message-passing operations between locations. The resulting architecture mimics time-then-space models that are typical to the spatio-temporal graph processing literature (Gao & Ribeiro, 2022). First, inputs are processed by the RNN$_{\text{emb}}$, as above, up to Eq. 13; then, a GC layer performs a convolution operation over the inter-location graph $G$:

$$\mathbf{H}_t' = \mathbf{A}_G \, \mathbf{H}_t \, \Theta + \mathbf{H}_t \, \Theta_{skip} + \mathbf{b} \tag{15}$$

where $\mathbf{H}_t = [\, \mathbf{H}_t^{fwd} \, || \, \mathbf{H}_t^{bwd} \,]$ and $\mathbf{A}_G$ is obtained from Eq. 2. Finally, RNN and GNN states are concatenated and fed to an MLP readout to produce the final imputation:

$$\hat{\mathbf{x}}_t[n] = \text{MLP}( \, [\, \mathbf{h}_t^{fwd}[n] \, || \, \mathbf{h}_t^{bwd}[n] \, || \, \mathbf{h}_t'[n] \, || \, \mathbf{e}[n] \,] \,) \tag{16}$$

**BRITS:** global bidirectional RNN model for generic MTS imputation from Cao et al. (2018) [2]. The 2D form is recovered by processing one MTS at a time, i.e., in practice, flattening the location dimension along the batch dimension.

**SAITS:** global self-attention-based MTS imputation model from Du et al. (2023) [3]. The 2D form is recovered by processing one MTS at a time, i.e., in practice, flattening the location dimension along the batch dimension.

**GRIN:** graph-based model for spatio-temporal data imputation from Cini et al. (2022) [4]. This baseline has access to the geographical coordinates, from which we compute the adjacency matrix with a thresholded Gaussian kernel (Shuman et al., 2013):

$$A_{ij} = A_{ji} = \begin{cases} \exp\left( -\dfrac{dist(i,j)^2}{\sigma^2} \right) & \text{if} \quad dist(i,j) < \delta \\ 0 & \text{otherwise} \end{cases} \tag{17}$$

---

[2] https://github.com/caow13/BRITS
[3] https://github.com/WenjieDu/SAITS
[4] https://github.com/Graph-Machine-Learning-Group/grin

with $dist(\cdot, \cdot)$ being the Haversine distance, $\sigma^2$ the distances variance, and $\delta$ a custom threshold. For all datasets with a number of nodes $N \geq 70$, we set $\delta$ to the required value to obtain an average of 10 edges per node, e.g., $\delta$(climatic datasets) $= 3500$ km. For smaller datasets, we fix $\delta$ to the corresponding value at $N = 70$. The **GRIN$_m$** variant adds self-loops to the original *Spatial Decoder* of GRIN.

## C  DETAILED EXPERIMENTAL SETTING

**Shared settings**    For GgNet and all baselines, we adopt the Adam optimizer (Kingma & Ba, 2015) with a learning rate $lr = 0.001$ paired with a cosine annealing learning rate scheduler. All models are trained for a maximum of 500 epochs, with a 30 epochs patience for early stopping. All batches have a size set to 32 and consider temporal windows of $t_w = 24$ time steps. As for the settings that are specific to virtual sensing: we set, for all methods, the probability of randomly masking training points to $p_{withen-points} = 0.05$, the probability of randomly masking entire channels to $p_{withen-channels} = 0.3$ and the weight of masked points during loss calculation to $w_{whiten} = 5$. Different values $w_{whiten} \in \{3, 5, 10\}$ led to a negligible difference in performance. A brief study on the robustness under changes in the $p_{withen-channels}$ mask can be found in Appendix E.6. Experiments are conducted within the Python (Van Rossum et al., 1995) library *Torch Spatiotemporal* (Cini & Marisca, 2022), which is also used to implement all methods. Experiments are tracked using *Weight and Biases* (Biewald, 2020). Code to reproduce the experiments is available online.[5]

**Hyperparameter settings on climatic dataset**    In the climatic data, all models are evaluated under different hyperparameter configurations, with increasing complexity. Separately for the daily and hourly datasets, the best-performing configuration of parameters on the validation set across the training epochs is used at testing. For GgNet, this resulted in the same set of parameters for both datasets: hidden size $d_h = 128$ for all convolutions and MLPs; location node embedding size $h_{e,G} = 16$, channel node embedding size $h_{e,g} = 8$; 2 blocks, each of which is composed of 3 layers of temporal convolution (with filter length $k_{\text{temp. filter}} = 3$ centered at each time step $t$), 1 layer of G-convolution and 1 layer of g-convolution; "*elu*" activation functions for the encoding MLPs, all graph convolutions and decoding MLPs; "*Tanh*" activation functions for the MLPs transforming the embeddings (Eq. 2). Hyperparameter choices for all baselines are available within the configurations file in the provided code. RNN variants are set to the same hidden size $h = 64$ for comparability.

**Hyperparameter settings on photovoltaic dataset**    In the photovoltaic dataset, to assess performance scaling with the number of locations, we set a common hidden size of $h = 64$ to all models and leave all other parameters to the best-performing setting in the climatic daily experiment. For all models, we also set a dropout value of $dr = 0.1$ in all experiments up to $N = 70$, and $dr = 0$ afterward.

### C.1  LOSS FUNCTIONS

**GgNet loss function**    As our loss function $\ell(\cdot, \cdot)$, we employ the sum of three quantile losses, (also referred to as 'pinball loss') (Koenker & Bassett Jr, 1978), with quantile levels selected as follows: $q_{-\sigma} = 0.159$, $q_\mu = 0.5$, and $q_{+\sigma} = 0.841$. The median quantile aligns with the Mean Absolute Error (MAE), while $q_{-\sigma}$ and $q_{+\sigma}$ allow for the concurrent estimation of uncertainties. We believe this is particularly relevant to virtual sensing, as not all channels in all locations can be equally reconstructed from the available data. As in (Cini et al., 2023a), further embedding regularizations can be considered in addition to the training objective.

**Baseline loss functions**    RNN and RNN$_{BiD}$ use MAE loss; RNN$_{emb}$ and RNN$_G$ use the same three-quantiles loss as GgNet; BRITS, SAITS and GRIN use same loss functions as the authors provide in the respective code implementations.

---

[5]https://github.com/gdefe/ggnet-virtual-sensing

## D    DATASETS DETAILS

**Climatic dataset:**    We obtain daily and hourly climatic data from the POWER Project's Daily and Hourly 2.3.5 version on 2023/02/26. This online database contains data about surface solar energy fluxes and other meteorological quantities obtained through satellite systems and further reanalysis. For radiation data, spatial resolution is $1.0°$ (latitude) by $1.0°$ (longitude); for meteorological data, this is $1/2°$ (latitude) by $5/8°$ (longitude). For each point on this global-scale grid, data are provided in time-series format with customizable temporal resolution. Further information, together with data and API, is available at the project website [6]. For daily data, we select the following 10 variables: *mean temperature* ($C$), *temperature range* ($C$), *maximum temperature* ($C$), *wind speed* ($m/s$), *relative humidity* (%), *precipitation* ($mm/$day), *dew/frost point* ($C$), *cloud amount* (%), *all-sky surface shortwave irradiance* ($W/m^2$) and *all-sky surface longwave irradiance* ($W/m^2$). Daily data extend for 30 years (1991-01-01 to 2022-12-31). For hourly data, we reduce the dimensionality to 7 variables, as *maximum temperature*, *temperature range* and *cloud amount* are not available at this resolution. Hourly data extend for 1 year (2022-01-01 to 2022-12-31).

**Photovoltaic dataset:**    We collect photovoltaic data from Hu et al. (2022), which simulate photovoltaic power production densely over the Continental North America. Simulations extend over the entire year 2019, with a temporal resolution of one hour, which we average across days to obtain daily data. The spatial resolution of the mesh grid is 12 km, resulting in 56,776 points, from which we randomly sample a few locations to recreate a sparse scenario. From the 13 simulated photovoltaic panels, we select module 00. Finally, we take the average across the 21 forecast members of the provided ensemble.

## E    ADDITIONAL ANALYSIS

### E.1    SCALABILITY AND COMPUTATION TIME

The asymptotic computational complexity of GgNet originates from the sum of the complexities of its components. The time and space complexity are both $\mathcal{O}(N^2TD) + \mathcal{O}(NTD^2)$. As it is often safe to assume that $D^2 \ll N^2$, the overall complexity reduces to $\mathcal{O}(N^2TD)$. The main bottleneck in GgNet is due to the inter-locations graph which makes message-passing operations scale quadratically w.r.t. the number of locations. If scalability to large $N$ is a concern, existing sparse graph learning methods can be considered (Niculae et al., 2023; Cini et al., 2023b). However, note that scenarios that involve significantly more sensors than the selected datasets would not likely be sparse and a different set of techniques should be used.

We also report here the computational times of different methods on the daily climatic dataset ($N = 235, T = 10958, D = 10$) under the best hyperparameter setting. It takes approximately (on average) 2h for BRITS to complete one training, 7h for SAITS, 16h for GgNet, 14h for GRIN. Approximate time per epoch are: 40s for BRITS, 4m 40s for SAITS, 6m for GgNet, 8m 30s for GRIN. The temporal modeling in GgNet is based on temporal convolutions, which makes it faster than the popular graph-based representative GRIN, which uses a recurrent model instead. As for BRITS and SAITS, these are faster than GgNet as they do not account for spatial dependencies. Timings are taken on a machine equipped with an NVIDIA A100 GPU.

### E.2    ABLATION STUDY

The GgNet architecture is composed of the following principal constituents: 3 layers of temporal convolutions, which we indicate with 3T, and one pairs of alternate spatial and channel convolutions: G - $g$. The sequence is iterated twice to allow for deeper processing, resulting in the following layers' pattern: 2(3T - G - $g$). We provide in Tab. 3 an ablation study by removing individual components; model's hidden size is $h = 64$, without residual connections and variable embeddings ($\mathbf{E}_g$). Considerably worse performances are obtained by removing the *g-convolution*, i.e., 2(3T - G), supporting the importance of modeling the relations with the covariates in reconstructing the missing channel in a sparse setting. Removing the temporal convolution, i.e., 2(G - $g$), forces the model to exchange only instantaneous information across locations and channels, missing the temporal

---

[6]https://power.larc.nasa.gov/

Table 3: Ablation study. Results are averaged across 3 runs. 2(3T - G - $g$) represents the architecture for which results are presented in the main text of the paper.

| CH. | TEMP. MEAN | TEMP. RANGE | TEMP. MAX | WIND SPEED | REL. HUM. | PREC. | TEMP. DEW | CLOUDS | IRR. SHORT | IRR. LONG | AVG (D) |
|---|---|---|---|---|---|---|---|---|---|---|---|
| 2(3T - G) | $9.2_{\pm0.5}$ | $25.9_{\pm3.0}$ | $8.4_{\pm1.2}$ | $33.1_{\pm1.1}$ | $8.5_{\pm0.4}$ | $83.0_{\pm3.7}$ | $14.8_{\pm1.6}$ | $33.1_{\pm0.6}$ | $17.7_{\pm1.5}$ | $4.5_{\pm0.1}$ | $23.8_{\pm0.7}$ |
| 2(G - $g$) | $4.8_{\pm0.8}$ | $18.5_{\pm2.4}$ | $3.9_{\pm1.0}$ | $28.0_{\pm2.4}$ | $5.6_{\pm0.8}$ | $69.1_{\pm0.7}$ | $7.6_{\pm1.3}$ | $20.6_{\pm1.9}$ | $11.3_{\pm1.2}$ | $3.2_{\pm0.2}$ | $17.3_{\pm1.1}$ |
| 2(3T - $g$) | $3.1_{\pm0.2}$ | $11.9_{\pm0.7}$ | $2.5_{\pm0.3}$ | $29.1_{\pm0.9}$ | $2.8_{\pm0.5}$ | $68.0_{\pm3.0}$ | $4.9_{\pm0.2}$ | $20.8_{\pm0.4}$ | $14.1_{\pm1.0}$ | $3.4_{\pm0.3}$ | $16.1_{\pm0.4}$ |
| 2(3T - G - $g$) | $2.6_{\pm0.2}$ | $10.9_{\pm1.7}$ | $2.6_{\pm0.3}$ | $25.0_{\pm1.4}$ | $2.8_{\pm0.3}$ | $61.7_{\pm1.4}$ | $4.4_{\pm0.5}$ | $17.3_{\pm0.4}$ | $9.2_{\pm0.8}$ | $3.2_{\pm0.2}$ | $14.0_{\pm0.1}$ |

context. Removing the *G-convolution*, i.e., 2(3T - $g$), negates the model from exploiting information from other locations at inference. This variant has access to the same information as other models that do not consider spatial information, e.g., SAITS.

### E.3 GEOGRAPHICALLY ISOLATED LOCATIONS

In this section, we qualitatively discuss the disadvantages of graph-based approaches in reconstructing virtual sensors in geographically sparse locations. This is done, in Fig. 6 (left), by an illustrative example on the daily climatic dataset. The plot compares the reconstruction of the *maximum tem-*

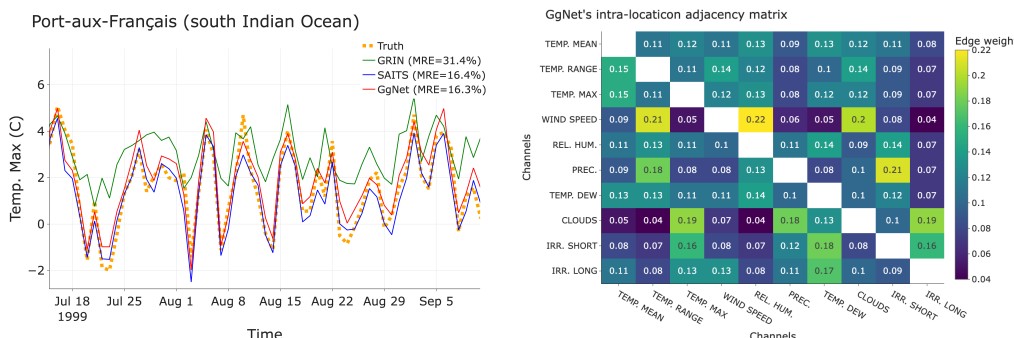

Figure 6: (Left) Reconstruction of *maximum temperature* (well correlated with other available variables) in Port-aux-Français, which cannot exploit observations at nearby locations as geographically isolated in the south Indian Ocean. (Right) colourmap visualization of the GgNet's learned intra-location graph ($g$), representation of the mutual dependencies between climatic channels.

*perature* channel, with different models, in Port-aux-Français, a geographically isolated location in the south Indian Ocean. For such a remote location, no information can be inferred from nearby points, making T → Y dependencies ineffective in performing virtual sensing. As a consequence, graph-based methods, e.g., GRIN, heavily underperform, as they rely on geographical proximity to build the graph and model spatial dependencies. On the contrary, GgNet or global recurrent models, e.g., SAITS, can effectively infer the missing channel by leveraging, at the target location, the learned dependencies between the target and the available covariates (C → Y dependencies). With the intent of enhancing this effect, we deliberately choose *maximum temperature* as target variable, which is highly correlated with other variables in the dataset.

### E.4 LEARNED DEPENDENCIES BETWEEN VARIABLES

In GgNet, relationships between channels are accounted for by means of the intra-location graph $g$. In particular, we learn the edge scores of its adjacency matrix from a D × D free parameter matrix, normalized with a softmax function. In Fig. 6 (right) we provide a colourmap visualization of the learned weighted (dense and bidirected) adjacency matrix. Interestingly, after processing the daily climatic dataset, several meaningful correlations have been captured: e.g., temperatures correlate with each other; *temperature range* affects *precipitation* and *wind speed*; *relative humidity* corre-

lates with temperatures, *precipitation* and *wind speed*; *precipitation* correlate with *clouds*; *longwave irradiation* correlates with clouds and *shortwave irradiation*.

### E.5    LOCATION-WISE ANALYSIS

Tables throughout the paper present results in a channel-wise fashion, i.e., for each channel $d$, performance metrics are averaged across test locations. In this additional analysis, we present, for the daily climatic dataset, results averaged across variables, i.e., location-wise results. With this anal-

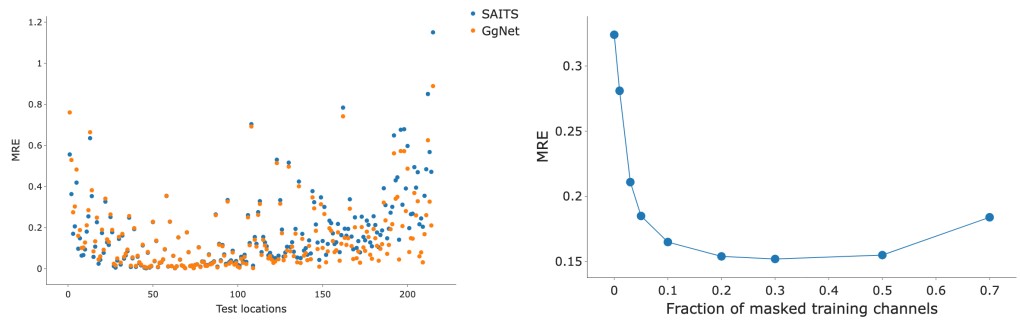

Figure 7: (left) location-wise analysis of GgNet performance over SAITS for the daily climatic dataset. (right) Robustness of GgNet to the fraction of masked training channels on the daily climatic dataset.

ysis, we aim to demonstrate that the improvement of GgNet over SAITS is not restricted to a few critical locations but is actually spread across the considered locations. As evidenced by Fig. 7 (left), GgNet improves over SAITS for $\sim 2/3$ of the considered target locations.

### E.6    ROBUSTNESS TESTS

With reference to Sec. 4.4, the training of GgNet and the adopted baselines is guided by means of a specific mask. In particular, we mask out a fraction of the available training channels (entirely for all timestamps in a batch) and train the models to reconstruct such masked portions of data. This is a core aspect, as it pushes the training toward learning the virtual-sensing task. In this regard, we investigate the accuracy of GgNet on the daily climatic dataset for different values of the fraction of masked training channels. The study is reported in Fig. 7 (right); the model's layer organization is 4T - 2(G-g), model's hidden size is $h = 64$, without residual connections and variable embeddings ($\mathbf{E}_g$). Results show a rapid improvement as soon as a small fraction is used, followed by robust performances.

### E.7    UNCERTAINTY ESTIMATES

Virtual sensing models are often employed to guide decision-making at unsampled locations. In this regard, it is important to pair the estimate for the virtual channel with a prediction of the corresponding uncertainty. In fact, channels at some locations may be, in principle, more challenging to reconstruct than others. In GgNet, uncertainty estimates are obtained by employing the sum of three quantile losses (Appendix C.1). Fig. 8 explores four different scenarios where the reconstruction is gradually more challenging. The extent of the uncertainty estimates is consistent with the dominant factors in assessing the quality of the predictions, i.e., the correlation between the target and the available covariates and the presence of similar latent representations in the training set.

### E.8    OTHER METRICS

Results are provided in the main text in terms of the most relevant metric for the task at hand. This is MRE for the climatic dataset, as different channels are expressed in different units, and MAE for the photovoltaic dataset, as power output is the only variable being reconstructed. Here, we also define

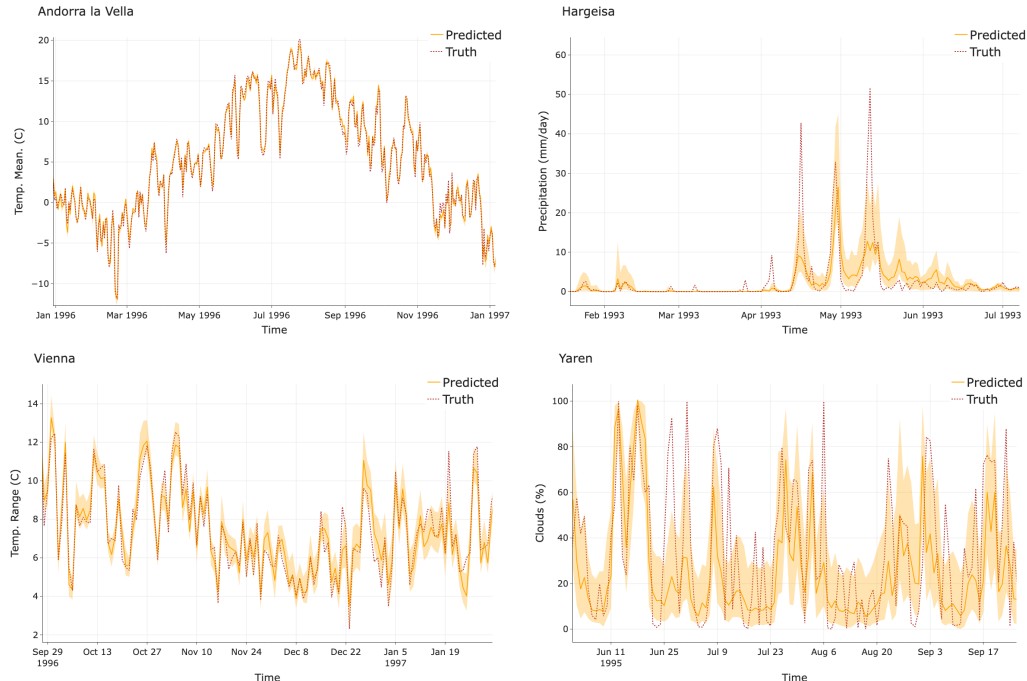

Figure 8: Daily climatic dataset: GgNet predictions for different channels at different locations, uncertainty estimates appear consistent with the difficulty of the reconstruction.

a custom metric, *Variance Rescaled Error* (VRE), tailored to this task:

$$\text{VRE}(\hat{\mathbf{x}}^d[n], \mathbf{x}^d[n]) = \frac{1}{T} \sum_{t=1}^{T} \frac{|\hat{x}_t^d[n] - x_t^d[n]|}{\sigma^d[n]} \tag{18}$$

where $\hat{\mathbf{x}}^d[n]$ and $\mathbf{x}^d[n]$ are the predicted and true $d$-th channel in $n$-th location, $\sigma^d[n]$ is the standard deviation along the temporal axis of the true signal. Results are consistent across different metrics. We support this statement by reporting below the testing accuracy, for all experiments, in terms of the other metrics. Specifically, Tab. 6 contains the channel-wise results for the hourly climatic dataset that were omitted from Tab. 1 to respect the space constraints.

Table 4: Daily climatic dataset: channel-wise MAE. Results are averaged across locations and 5 random seeds. The best-performing method is in bold, the second-best is underlined.

| CH. | TEMP. MEAN | TEMP. RANGE | TEMP. MAX | WIND SPEED | REL. HUM. | PREC. | TEMP. DEW | CLOUDS | IRR. SHORT | IRR. LONG |
|---|---|---|---|---|---|---|---|---|---|---|
| KNN | $3.21_{\pm0.39}$ | $2.84_{\pm0.26}$ | $2.97_{\pm0.35}$ | $1.47_{\pm0.19}$ | $9.00_{\pm0.30}$ | $2.93_{\pm0.27}$ | $3.49_{\pm0.39}$ | $19.53_{\pm0.33}$ | $36.46_{\pm1.02}$ | $21.87_{\pm3.23}$ |
| BRITS | $0.82_{\pm0.09}$ | $1.19_{\pm0.16}$ | $0.78_{\pm0.12}$ | $1.40_{\pm0.04}$ | $4.04_{\pm0.64}$ | $2.30_{\pm0.21}$ | $1.22_{\pm0.15}$ | $17.16_{\pm0.65}$ | $42.41_{\pm2.18}$ | $12.49_{\pm1.05}$ |
| GRIN | $0.86_{\pm0.10}$ | $1.26_{\pm0.14}$ | $1.04_{\pm0.11}$ | $1.33_{\pm0.06}$ | $4.15_{\pm0.48}$ | $2.20_{\pm0.16}$ | $1.26_{\pm0.12}$ | $16.70_{\pm0.32}$ | $31.14_{\pm1.22}$ | $14.30_{\pm0.92}$ |
| GRIN$_m$ | $0.59_{\pm0.16}$ | $0.88_{\pm0.09}$ | $0.67_{\pm0.12}$ | $1.33_{\pm0.12}$ | $2.64_{\pm0.65}$ | $2.00_{\pm0.19}$ | $0.93_{\pm0.14}$ | $\underline{11.79}_{\pm0.39}$ | $\underline{23.70}_{\pm0.94}$ | $13.13_{\pm1.22}$ |
| SAITS | $\underline{0.49}_{\pm0.07}$ | $\underline{0.76}_{\pm0.10}$ | $\underline{0.56}_{\pm0.10}$ | $\underline{1.17}_{\pm0.05}$ | $\underline{2.42}_{\pm0.75}$ | $\underline{1.88}_{\pm0.20}$ | $\underline{0.73}_{\pm0.14}$ | $11.86_{\pm0.38}$ | $28.80_{\pm1.58}$ | $\mathbf{10.07}_{\pm0.78}$ |
| **GGNET** | $\mathbf{0.42}_{\pm0.07}$ | $\mathbf{0.65}_{\pm0.02}$ | $\mathbf{0.48}_{\pm0.06}$ | $\mathbf{1.04}_{\pm0.08}$ | $\mathbf{1.98}_{\pm0.55}$ | $\mathbf{1.73}_{\pm0.13}$ | $\mathbf{0.62}_{\pm0.12}$ | $\mathbf{9.48}_{\pm0.35}$ | $\mathbf{18.73}_{\pm1.42}$ | $\underline{10.39}_{\pm0.81}$ |
| RNN | $1.16_{\pm0.10}$ | $1.49_{\pm0.13}$ | $1.40_{\pm0.12}$ | $1.42_{\pm0.05}$ | $5.08_{\pm0.50}$ | $2.39_{\pm0.20}$ | $1.57_{\pm0.14}$ | $20.38_{\pm0.24}$ | $44.15_{\pm1.02}$ | $13.97_{\pm1.17}$ |
| RNN$_{bi}$ | $0.87_{\pm0.09}$ | $1.30_{\pm0.14}$ | $1.12_{\pm0.10}$ | $1.35_{\pm0.06}$ | $4.34_{\pm0.47}$ | $2.22_{\pm0.16}$ | $1.25_{\pm0.13}$ | $18.06_{\pm0.19}$ | $38.59_{\pm0.90}$ | $13.13_{\pm1.21}$ |
| RNN$_{emb}$ | $0.85_{\pm0.07}$ | $1.19_{\pm0.12}$ | $1.03_{\pm0.08}$ | $1.31_{\pm0.07}$ | $4.01_{\pm0.46}$ | $2.20_{\pm0.18}$ | $1.23_{\pm0.13}$ | $17.36_{\pm0.26}$ | $35.48_{\pm1.18}$ | $13.18_{\pm0.71}$ |
| RNN$_G$ | $0.74_{\pm0.04}$ | $1.12_{\pm0.12}$ | $0.94_{\pm0.07}$ | $1.20_{\pm0.06}$ | $3.80_{\pm0.30}$ | $2.13_{\pm0.20}$ | $1.06_{\pm0.11}$ | $15.90_{\pm0.33}$ | $28.47_{\pm1.09}$ | $13.04_{\pm1.13}$ |

Table 5: Daily climatic dataset: channel-wise VRE (%). Results are averaged across locations and 5 random seeds. The best-performing method is in bold, the second-best is underlined.

| Ch. | Temp. Mean | Temp. Range | Temp. Max | Wind Speed | Rel. Hum. | Prec. | Temp. Dew | Clouds | Irr. Short | Irr. Long | Avg (D) |
|---|---|---|---|---|---|---|---|---|---|---|---|
| KNN | $105.8_{\pm15.3}$ | $213.3_{\pm55.6}$ | $90.9_{\pm7.4}$ | $116.4_{\pm26.3}$ | $102.3_{\pm3.4}$ | $56.8_{\pm4.2}$ | $100.8_{\pm10.0}$ | $72.1_{\pm2.7}$ | $59.7_{\pm3.7}$ | $95.9_{\pm14.8}$ | $101.4_{\pm9.3}$ |
| BRITS | $27.5_{\pm3.1}$ | $75.1_{\pm11.2}$ | $23.4_{\pm3.4}$ | $104.0_{\pm13.2}$ | $45.2_{\pm6.6}$ | $41.6_{\pm1.2}$ | $30.9_{\pm4.1}$ | $63.7_{\pm3.4}$ | $67.2_{\pm4.4}$ | $53.9_{\pm5.2}$ | $53.2_{\pm1.9}$ |
| GRIN | $29.7_{\pm3.6}$ | $75.8_{\pm12.2}$ | $31.0_{\pm2.5}$ | $99.4_{\pm13.7}$ | $48.2_{\pm4.1}$ | $39.9_{\pm1.4}$ | $36.1_{\pm2.1}$ | $61.5_{\pm1.6}$ | $50.4_{\pm3.0}$ | $64.3_{\pm9.5}$ | $53.6_{\pm3.2}$ |
| $GRIN_m$ | $23.3_{\pm5.1}$ | $55.6_{\pm7.3}$ | $21.9_{\pm5.2}$ | $101.8_{\pm18.5}$ | $30.1_{\pm6.6}$ | $37.0_{\pm2.8}$ | $27.4_{\pm3.1}$ | $\underline{43.6}_{\pm1.8}$ | $\underline{39.6}_{\pm3.3}$ | $64.8_{\pm10.2}$ | $44.5_{\pm2.7}$ |
| SAITS | $\underline{16.8}_{\pm1.4}$ | $\underline{44.8}_{\pm5.5}$ | $\underline{16.5}_{\pm2.9}$ | $\underline{88.8}_{\pm8.9}$ | $\underline{26.0}_{\pm7.4}$ | $\underline{33.7}_{\pm1.6}$ | $\underline{20.1}_{\pm3.2}$ | $44.0_{\pm2.1}$ | $47.5_{\pm5.4}$ | $\mathbf{45.2}_{\pm5.7}$ | $38.3_{\pm1.5}$ |
| **GGNet** | $\mathbf{14.6}_{\pm3.2}$ | $\mathbf{35.8}_{\pm3.0}$ | $\mathbf{14.4}_{\pm1.8}$ | $\mathbf{80.5}_{\pm14.5}$ | $\mathbf{22.1}_{\pm5.3}$ | $\mathbf{30.9}_{\pm1.3}$ | $\mathbf{17.3}_{\pm2.2}$ | $\mathbf{35.2}_{\pm1.7}$ | $\mathbf{31.3}_{\pm3.8}$ | $\underline{48.9}_{\pm6.3}$ | $\mathbf{33.1}_{\pm1.9}$ |
| RNN | $37.2_{\pm4.1}$ | $89.6_{\pm5.4}$ | $39.3_{\pm3.6}$ | $105.2_{\pm12.0}$ | $56.8_{\pm4.3}$ | $44.0_{\pm2.2}$ | $42.0_{\pm2.6}$ | $74.7_{\pm1.1}$ | $69.1_{\pm3.3}$ | $60.0_{\pm7.0}$ | $61.8_{\pm1.1}$ |
| $RNN_{bi}$ | $28.9_{\pm3.4}$ | $74.8_{\pm9.8}$ | $32.7_{\pm3.3}$ | $101.0_{\pm11.6}$ | $48.7_{\pm4.2}$ | $40.9_{\pm2.2}$ | $33.3_{\pm2.0}$ | $66.4_{\pm1.8}$ | $61.4_{\pm3.9}$ | $57.3_{\pm7.5}$ | $54.5_{\pm0.8}$ |
| $RNN_{emb}$ | $26.9_{\pm4.1}$ | $63.2_{\pm3.8}$ | $29.7_{\pm3.0}$ | $99.8_{\pm12.7}$ | $45.8_{\pm3.9}$ | $40.0_{\pm1.4}$ | $33.3_{\pm3.0}$ | $63.8_{\pm1.4}$ | $57.3_{\pm4.6}$ | $58.4_{\pm7.8}$ | $51.8_{\pm2.3}$ |
| $RNN_G$ | $24.5_{\pm3.2}$ | $60.2_{\pm2.5}$ | $28.0_{\pm2.3}$ | $93.1_{\pm11.8}$ | $43.6_{\pm2.4}$ | $38.6_{\pm2.0}$ | $29.0_{\pm2.1}$ | $58.5_{\pm1.5}$ | $46.2_{\pm3.3}$ | $58.6_{\pm10.5}$ | $48.0_{\pm2.0}$ |

Table 6: Hourly climatic dataset: channel-wise MRE (%). Results are averaged across locations and 5 random seeds. The best-performing method is in bold, the second-best is underlined.

| Ch. | Temp. Mean | Wind Speed | Rel. Hum. | Prec. | Temp. Dew | Irr. Short | Irr. Long | Avg (H) |
|---|---|---|---|---|---|---|---|---|
| KNN | $15.9_{\pm2.4}$ | $38.0_{\pm5.2}$ | $14.6_{\pm1.8}$ | $133.6_{\pm7.8}$ | $23.1_{\pm3.4}$ | $21.4_{\pm1.9}$ | $6.7_{\pm0.4}$ | $36.2_{\pm1.9}$ |
| BRITS | $7.4_{\pm2.5}$ | $34.6_{\pm1.4}$ | $6.5_{\pm1.2}$ | $85.4_{\pm1.1}$ | $9.1_{\pm1.5}$ | $36.5_{\pm2.0}$ | $4.6_{\pm0.4}$ | $26.3_{\pm0.8}$ |
| GRIN | $6.5_{\pm2.3}$ | $35.5_{\pm1.0}$ | $5.3_{\pm1.1}$ | $86.9_{\pm2.3}$ | $8.6_{\pm1.2}$ | $\mathbf{16.2}_{\pm1.0}$ | $4.4_{\pm0.3}$ | $23.3_{\pm0.7}$ |
| $GRIN_m$ | $6.0_{\pm2.2}$ | $34.5_{\pm2.1}$ | $4.8_{\pm0.9}$ | $85.5_{\pm1.1}$ | $8.0_{\pm1.4}$ | $\mathbf{16.1}_{\pm0.8}$ | $4.3_{\pm0.3}$ | $22.7_{\pm0.4}$ |
| SAITS | $\underline{4.7}_{\pm1.8}$ | $\mathbf{29.0}_{\pm0.8}$ | $\mathbf{4.3}_{\pm0.8}$ | $\underline{82.8}_{\pm1.2}$ | $\underline{6.6}_{\pm1.2}$ | $24.4_{\pm1.6}$ | $\underline{3.7}_{\pm0.2}$ | $\underline{22.2}_{\pm0.5}$ |
| **GGNet** | $\mathbf{4.5}_{\pm1.9}$ | $\underline{29.2}_{\pm2.5}$ | $\mathbf{4.3}_{\pm0.7}$ | $\mathbf{78.4}_{\pm1.1}$ | $\mathbf{5.7}_{\pm0.9}$ | $\underline{17.4}_{\pm3.8}$ | $\mathbf{3.6}_{\pm0.2}$ | $\mathbf{20.4}_{\pm0.6}$ |
| RNN | $8.0_{\pm2.7}$ | $38.9_{\pm1.3}$ | $6.6_{\pm0.8}$ | $90.8_{\pm1.3}$ | $9.7_{\pm1.5}$ | $43.2_{\pm3.3}$ | $4.5_{\pm0.3}$ | $28.8_{\pm0.8}$ |
| $RNN_{bi}$ | $5.9_{\pm2.0}$ | $36.5_{\pm1.5}$ | $5.3_{\pm0.9}$ | $87.1_{\pm1.3}$ | $7.7_{\pm1.2}$ | $33.3_{\pm2.0}$ | $4.1_{\pm0.3}$ | $25.7_{\pm0.5}$ |
| $RNN_{emb}$ | $6.6_{\pm2.8}$ | $36.4_{\pm0.9}$ | $5.6_{\pm1.1}$ | $86.8_{\pm1.1}$ | $8.2_{\pm1.5}$ | $35.6_{\pm3.2}$ | $4.5_{\pm0.4}$ | $26.2_{\pm0.9}$ |
| $RNN_G$ | $5.7_{\pm2.4}$ | $32.2_{\pm2.7}$ | $5.1_{\pm0.9}$ | $83.0_{\pm3.0}$ | $7.1_{\pm1.0}$ | $25.1_{\pm2.7}$ | $4.0_{\pm0.4}$ | $23.2_{\pm1.1}$ |

Table 7: Hourly climatic dataset: channel-wise MAE. Results are averaged across locations and 5 random seeds. The best-performing method is in bold, the second-best is underlined.

| Ch. | Temp. Mean | Wind Speed | Rel. Hum. | Prec. | Temp. Dew | Irr. Short | Irr. Long |
|---|---|---|---|---|---|---|---|
| KNN | $3.24_{\pm0.29}$ | $1.64_{\pm0.25}$ | $10.59_{\pm1.05}$ | $0.16_{\pm0.01}$ | $3.54_{\pm0.49}$ | $43.52_{\pm4.19}$ | $24.54_{\pm1.55}$ |
| BRITS | $1.49_{\pm0.44}$ | $1.49_{\pm0.13}$ | $4.68_{\pm0.76}$ | $\underline{0.10}_{\pm0.01}$ | $1.39_{\pm0.24}$ | $74.13_{\pm4.00}$ | $16.84_{\pm1.39}$ |
| GRIN | $1.32_{\pm0.42}$ | $1.53_{\pm0.09}$ | $3.84_{\pm0.74}$ | $\underline{0.10}_{\pm0.01}$ | $1.31_{\pm0.20}$ | $\mathbf{32.91}_{\pm2.22}$ | $15.99_{\pm1.11}$ |
| $GRIN_m$ | $1.23_{\pm0.40}$ | $1.49_{\pm0.16}$ | $3.47_{\pm0.60}$ | $\underline{0.10}_{\pm0.00}$ | $1.22_{\pm0.23}$ | $\mathbf{32.74}_{\pm1.75}$ | $15.60_{\pm0.99}$ |
| SAITS | $\underline{0.96}_{\pm0.32}$ | $\mathbf{1.25}_{\pm0.08}$ | $\mathbf{3.11}_{\pm0.50}$ | $\underline{0.10}_{\pm0.01}$ | $\underline{1.00}_{\pm0.18}$ | $49.67_{\pm3.34}$ | $\underline{13.31}_{\pm0.77}$ |
| **GGNet** | $\mathbf{0.91}_{\pm0.35}$ | $\underline{1.26}_{\pm0.14}$ | $\mathbf{3.11}_{\pm0.47}$ | $\mathbf{0.09}_{\pm0.01}$ | $\mathbf{0.87}_{\pm0.15}$ | $\underline{35.28}_{\pm7.33}$ | $\mathbf{13.14}_{\pm0.49}$ |
| RNN | $1.61_{\pm0.48}$ | $1.68_{\pm0.12}$ | $4.76_{\pm0.50}$ | $0.11_{\pm0.00}$ | $1.49_{\pm0.23}$ | $87.86_{\pm6.70}$ | $16.21_{\pm1.09}$ |
| $RNN_{bi}$ | $1.19_{\pm0.37}$ | $1.57_{\pm0.13}$ | $3.82_{\pm0.59}$ | $0.10_{\pm0.01}$ | $1.18_{\pm0.18}$ | $67.75_{\pm4.15}$ | $15.00_{\pm1.08}$ |
| $RNN_{emb}$ | $1.34_{\pm0.51}$ | $1.57_{\pm0.09}$ | $4.09_{\pm0.71}$ | $0.10_{\pm0.01}$ | $1.25_{\pm0.24}$ | $72.50_{\pm7.11}$ | $16.29_{\pm1.30}$ |
| $RNN_G$ | $1.14_{\pm0.43}$ | $1.39_{\pm0.14}$ | $3.72_{\pm0.62}$ | $0.10_{\pm0.00}$ | $1.09_{\pm0.17}$ | $50.99_{\pm5.59}$ | $14.71_{\pm1.28}$ |

Table 8: Hourly climatic dataset: channel-wise VRE (%). Results are averaged across locations and 5 random seeds. The best-performing method is in bold, the second-best is underlined.

| Ch. | Temp. Mean | Wind Speed | Rel. Hum. | Prec. | Temp. Dew | Irr. Short | Irr. Long | Avg (H) |
|---|---|---|---|---|---|---|---|---|
| KNN | $77.4_{\pm5.2}$ | $96.9_{\pm13.9}$ | $83.9_{\pm11.9}$ | $49.1_{\pm6.6}$ | $106.9_{\pm21.2}$ | $16.3_{\pm1.4}$ | $83.5_{\pm8.3}$ | $73.4_{\pm4.1}$ |
| BRITS | $35.9_{\pm7.0}$ | $83.3_{\pm4.4}$ | $36.1_{\pm5.8}$ | $29.0_{\pm0.8}$ | $38.4_{\pm5.0}$ | $26.8_{\pm1.5}$ | $59.2_{\pm6.5}$ | $44.1_{\pm2.9}$ |
| GRIN | $37.8_{\pm7.1}$ | $85.0_{\pm2.2}$ | $32.6_{\pm8.2}$ | $29.2_{\pm1.1}$ | $42.7_{\pm8.6}$ | $\mathbf{12.2}_{\pm0.8}$ | $59.1_{\pm4.3}$ | $42.7_{\pm1.1}$ |
| GRIN$_m$ | $40.1_{\pm8.3}$ | $82.8_{\pm6.3}$ | $30.0_{\pm5.5}$ | $29.2_{\pm0.4}$ | $40.0_{\pm8.3}$ | $12.1_{\pm0.6}$ | $58.2_{\pm7.4}$ | $41.8_{\pm2.7}$ |
| SAITS | $\underline{24.3}_{\pm4.3}$ | $\mathbf{69.9}_{\pm4.0}$ | $\mathbf{25.5}_{\pm3.5}$ | $\underline{28.3}_{\pm0.6}$ | $\underline{28.1}_{\pm3.7}$ | $18.3_{\pm1.3}$ | $\mathbf{46.5}_{\pm1.1}$ | $\underline{34.4}_{\pm1.3}$ |
| **GGNet** | $\mathbf{24.0}_{\pm6.0}$ | $\underline{70.6}_{\pm2.8}$ | $\underline{26.3}_{\pm5.2}$ | $\mathbf{27.3}_{\pm1.4}$ | $\mathbf{25.3}_{\pm3.1}$ | $\underline{12.9}_{\pm2.5}$ | $\underline{46.9}_{\pm4.4}$ | $\mathbf{33.3}_{\pm0.9}$ |
| RNN | $40.5_{\pm7.2}$ | $93.5_{\pm4.7}$ | $36.5_{\pm4.2}$ | $31.2_{\pm0.9}$ | $44.3_{\pm4.5}$ | $31.8_{\pm2.3}$ | $56.0_{\pm2.0}$ | $47.7_{\pm2.0}$ |
| RNN$_{bi}$ | $29.9_{\pm5.9}$ | $87.4_{\pm4.3}$ | $30.5_{\pm5.4}$ | $29.5_{\pm1.0}$ | $33.3_{\pm3.1}$ | $24.8_{\pm1.4}$ | $51.7_{\pm2.2}$ | $41.0_{\pm1.3}$ |
| RNN$_{emb}$ | $35.6_{\pm8.4}$ | $88.4_{\pm3.2}$ | $33.8_{\pm6.9}$ | $29.1_{\pm1.0}$ | $33.9_{\pm2.9}$ | $26.2_{\pm2.5}$ | $57.2_{\pm1.8}$ | $43.4_{\pm2.2}$ |
| RNN$_G$ | $32.9_{\pm11.1}$ | $78.0_{\pm4.6}$ | $31.0_{\pm6.6}$ | $27.8_{\pm1.6}$ | $33.7_{\pm5.0}$ | $18.5_{\pm1.8}$ | $52.9_{\pm2.2}$ | $39.3_{\pm2.7}$ |

Table 9: Photovoltaic dataset: MRE (%). For each N, results are averaged across 5 sampling of the locations and 5 different runs for each set (25 runs in total). The best-performing method is in bold, the second-best is underlined.

| N$_{\text{LOCATIONS}}$ | 10 | 20 | 30 | 50 | 70 | 100 | 150 | 200 |
|---|---|---|---|---|---|---|---|---|
| GRIN | $20.1 \pm 6.3$ | $18.8 \pm 2.7$ | $18.6 \pm 2.2$ | $18.2 \pm 2.2$ | $18.8 \pm 1.8$ | $18.4 \pm 1.4$ | $18.3 \pm 1.7$ | $18.5 \pm 1.6$ |
| GRIN$_m$ | $17.2 \pm 4.7$ | $16.0 \pm 2.4$ | $\underline{15.6} \pm 2.1$ | $\underline{15.0} \pm 1.5$ | $\underline{15.1} \pm 1.2$ | $14.8 \pm 0.8$ | $\underline{14.6} \pm 0.7$ | $\underline{14.6} \pm 0.6$ |
| SAITS | $\underline{16.0} \pm 3.0$ | $\mathbf{15.8} \pm 2.3$ | $\underline{15.6} \pm 1.8$ | $15.1 \pm 1.4$ | $15.4 \pm 1.3$ | $\underline{14.7} \pm 1.0$ | $\underline{14.6} \pm 0.7$ | $\underline{14.6} \pm 0.7$ |
| **GGNet** | $\mathbf{15.6} \pm 3.5$ | $\underline{15.9} \pm 2.0$ | $\mathbf{15.3} \pm 1.9$ | $\mathbf{14.7} \pm 1.5$ | $\mathbf{14.5} \pm 1.3$ | $\mathbf{13.7} \pm 1.0$ | $\mathbf{12.8} \pm 0.7$ | $\mathbf{12.5} \pm 0.7$ |

Table 10: Photovoltaic dataset: VRE (%). For each N, results are averaged across 5 sampling of the locations and 5 different runs for each set (25 runs in total). The best-performing method is in bold, the second-best is underlined.

| N$_{\text{LOCATIONS}}$ | 10 | 20 | 30 | 50 | 70 | 100 | 150 | 200 |
|---|---|---|---|---|---|---|---|---|
| GRIN | $40.7 \pm 11.9$ | $38.2 \pm 4.3$ | $37.8 \pm 3.6$ | $36.0 \pm 2.5$ | $37.4 \pm 2.7$ | $37.0 \pm 2.1$ | $36.9 \pm 2.8$ | $37.4 \pm 3.0$ |
| GRIN$_m$ | $34.8 \pm 7.1$ | $32.8 \pm 4.1$ | $\underline{31.7} \pm 3.3$ | $\underline{29.8} \pm 1.8$ | $\underline{30.3} \pm 1.7$ | $30.0 \pm 1.2$ | $\underline{29.5} \pm 1.2$ | $29.8 \pm 1.2$ |
| SAITS | $\underline{32.8} \pm 4.6$ | $\mathbf{32.4} \pm 3.8$ | $\underline{31.7} \pm 2.6$ | $30.0 \pm 2.0$ | $30.8 \pm 1.9$ | $\underline{29.7} \pm 1.3$ | $\underline{29.5} \pm 1.2$ | $\underline{29.6} \pm 1.3$ |
| **GGNet** | $\mathbf{31.8} \pm 4.4$ | $\underline{32.6} \pm 3.4$ | $\mathbf{31.2} \pm 2.6$ | $\mathbf{29.2} \pm 2.0$ | $\mathbf{29.0} \pm 2.0$ | $\mathbf{27.8} \pm 1.7$ | $\mathbf{26.0} \pm 1.4$ | $\mathbf{25.5} \pm 1.4$ |

