# OpenReview forum: "Graph-based Virtual Sensing from Sparse and Partial Multivariate Observations"
_ICLR.cc/2024/Conference — ICLR 2024 poster_

### Official Review · Reviewer_MbBb · 2023-10-25

**Soundness:** 3 good
**Presentation:** 3 good
**Contribution:** 3 good
**Rating:** 5
**Confidence:** 4

**Summary:**

This paper aims to address the data sparsity problem by using multivariate data. The authors proposed a model GgNet which can leverage the correlations/dependencies between the target variable/channel and the covariates to reconstruct the target values at certain locations. Extensive experiments are conducted.

**Strengths:**

1. The paper is well-presented and well-organized.
2. The paper proposed a new model GgNet aiming to reconstruct the target values at certain locations.
3. Extensive experiments are conducted to validate the proposed model, and the results seem promising.

**Weaknesses:**

1. This paper has a lot of assumptions, for example, it assumes the covariates are always available, and the mutual dependencies between target and covariates are invariant and can be leveraged at all locations. Here are some concerns: we cannot guarantee that those assumptions hold for all cases (at least, there should be more examples or references to support them), and if they do not hold, does the proposed method still work?
2. It seems the method requires the latent representations of locations where the target values need to be reconstructed are close, so is it possible to deal with the cases where the latent representations of locations are remote?
3. The model somewhat lacks novelty, it seems the final model just combines several existing models.

**Questions:**

Please address the questions above.

---

> ### Author Response · Authors · 2023-11-17
>
> We thank the reviewer for the constructive review and feedback.
> As for the raised concerns and questions, individual points are addressed below.
>
>
> **W1. Paper's assumptions**\
> Please allow us to clarify a possible misunderstanding. Our assumptions are only those dictated by the task which our method has been specifically designed to solve. In particular, for what concerns the availability of covariates, at least some of these are needed for the task to make sense, otherwise, no reconstruction would be possible in sparse settings.
> Furthermore, note that we do *not* require all covariates to be available at all locations. On the contrary, as shown in Fig.1, covariates can be partially missing at any location.
> Clearly, the quality of the reconstruction will depend on the amount of relevant information available to condition the virtual sensing model.
> We would like to point out that the covariate availability assumption is not a weakness of our proposed approach, but a hard requirement of the addressed sparse virtual sensing task. In general, this corresponds to the necessity of all ML models to condition inference on observations.
>
> Then, similarly to the previous point, assuming a mutual dependency between the target and covariates is a requirement imposed by the task. If this cannot be leveraged at the target location, it is not possible to perform any inference in a sparse setting.
>
> Finally, assuming synchronous observations and overlapping time frames among sensors is necessary to exploit the spatial dimension.
> Although the synchronicity assumption could be alleviated to some extent, we consider this aspect orthogonal to those we deal with in the present paper and mention this direction as a possible future work.
>
>
>
> **W2. Far-apart latent representation of locations**\
> In our framework, far-apart representations in the latent space can, potentially, translate into widely different observed dynamics. In the transductive setting, the masking strategy still brings supervision for training locations with isolated representations. However, in general, if no available observed location is at all similar to the target one, there may be no guarantee of the accuracy of the reconstruction. This is due to the intrinsic limits of any data-driven method used to extrapolate beyond the training conditions.
>
>
>
> **W3. Discussion on the paper's novelty**\
> The focus of the paper is not on introducing new layers or new neural operators.
> Instead, we are the first to explore (multivariate) virtual sensing for sparse settings, which is relevant in many practical applications.
> Most importantly, we introduce a conceptualization and propose a methodology to address the general problem, which goes far beyond the possible specific implementations. Nevertheless, we also propose a possible implementation (GgNet) of such a methodology, which utilizes an original nested graph representation for effectively modeling dependencies in such a context.
> We finally provide experimental evidence of how the proposed model can overcome the limitations of previous approaches.
> We believe these claims are significant and contribute to advancing the field in a meaningful way.
> We refer the reviewer to the introduction of the paper where our core novel contributions are explicitly stated. After the revision, we have added references to the section supporting each claim.
> We hope this clarifies any doubts regarding the novelty aspect of the paper; if not, we would deeply appreciate any further comment.
>
> We hope to have clarified all of the reviewer’s concerns and are happy to provide further details.

---

> > ### Comment · Reviewer_MbBb · 2023-11-22
> >
> > Thanks for the reply! Most of my concerns have been addressed.

---

> > > ### Author Response · Authors · 2023-11-22
> > >
> > > Thank you again. We remain open to address any additional questions you may have. We also hope that our responses have provided sufficient information for you to consider recommending acceptance.

---

### Official Review · Reviewer_cynR · 2023-10-30

**Soundness:** 3 good
**Presentation:** 2 fair
**Contribution:** 3 good
**Rating:** 6
**Confidence:** 4

**Summary:**

The paper presents a novel framework for virtual sensing of missing multivariate spatio-temporal data in settings with limited sensor coverage. The authors introduce GgNet, a deep learning architecture capable of leveraging inter-location dependencies and relationships between covariates for accurate data reconstruction.
The contributions of the paper include:
•	The introduction of a novel deep learning architecture designed for multivariate virtual sensing tasks, particularly in sparse scenarios.
•	Nested Graph Structure: GgNet employs a nested graph structure to capture both spatial relationships between locations and dependencies between covariates, learning them end-to-end for improved reconstruction accuracy.

**Strengths:**

•	The problem formulation of multivariate virtual sensing in sparse scenarios is itself an original and significant contribution, as it addresses a practical challenge across various domains. The introduction of GgNet leverages a nested graph structure to capture dependencies between covariates and relations across locations, is highly original. The combination of graph-based modeling and deep learning techniques to address the problem is innovative and distinguishes this work from prior methods in the field.
•	The technical content, methodology, and experimental design are well-supported with empirical evidence. The thoroughness of the experimental evaluation, conducted across multiple datasets, with different degrees of sparsity and temporal resolutions, enhances the paper's quality. The extensive experimentation, use of real-world datasets, and evaluation further emphasize the paper's significance.

**Weaknesses:**

•	The paper, while generally well-structured, could benefit from more explicit and detailed explanations in certain technical aspects. For instance, the paper could provide a more comprehensive explanation of the nested graph structure used in GgNet, which is a key component of the proposed method.

**Questions:**

•	The paper mentions the computational time for GgNet and other baselines but does not discuss scalability. How does GgNet's performance scale with more extensive sensor networks or larger datasets?

---

> ### Author Response · Authors · 2023-11-17
>
> We thank the reviewer for the review and positive feedback.
> As for the raised concerns and questions, individual points are addressed below.
>
>
> **W1. More detailed explanations in certain technical aspects**\
> We understand that there are several important technical details that were deferred to the appendix (in particular, in Appendix B). However, due to space constraints, we preferred to allocate more space to the conceptualization of the problem and to the methodological aspects that are the main paper contributions.
>
>
>
> **Q1. Discussion on scalability**\
> Scalability is discussed in Appendix E.1. The computational complexity of GgNet originates from the sum of the complexities of its components.
> The time and space complexity are both $\mathcal O(N^2TD) + \mathcal O(NTD^2)$.
> As it is often safe to assume that $D^2 \ll N^2$, the overall complexity reduces to $\mathcal O(N^2TD)$.
> If scalability to large N is found to be a concern, existing sparse graph learning alternative methods, e.g., [1], can be integrated into the approach.
> We thank the reviewer for the question and have expanded the discussion on scalability in Appendix E.1 of the revision.
>
> We hope to have clarified all of the reviewer’s concerns and are happy to provide further details.
>
> **References:**\
> [1] Niculae et al., Discrete latent structure in neural networks. arXiv preprint arXiv:2301.07473, 2023.

---

### Official Review · Reviewer_L9oj · 2023-10-31

**Soundness:** 3 good
**Presentation:** 3 good
**Contribution:** 3 good
**Rating:** 8
**Confidence:** 2

**Summary:**

In this paper, the authors presented a novel graph-based framework for virtual sensing from sparse multivariate spatio-temporal observations, called GgNet. They leveraged a nested graph structure to account for dependencies between covariates and relations across locations and learned end-to-end to maximize reconstruction accuracy. The GgNet achieves superior performance in settings with poor sensor coverage, where other state of the art fail, and contributes to the methodological advancement of the field as well as a powerful tool in practical applications.

**Strengths:**

①	The GgNet proposed in the article effectively addresses the limitations of traditional methods in the context of sparse sensor coverage by inferring signal values at unmonitored locations through learning dependencies between variables and positions.
②	The article provides numerous details in the appendices to help readers gain a more comprehensive understanding of the framework. For instance, Appendix B and C offered a detailed description of the experimental settings and baseline parameter details.
③	The author employed effective training method. They masked an additional small fraction of data points, at random, in the training channels, which can enforce robustness to random missing value.
④	The experimental baseline selection is quite comprehensive and reasonable, including KNN, BRITS, SAITS, and GRIN, along with various progressively more advanced recurrent RNN. These choices in baselines contribute to a more comprehensive and reliable set of experimental results.

**Weaknesses:**

①	This article lacks a thorough analysis of the experimental results and a comprehensive interpretation of the data. It primarily presents the experimental outcomes in tables without providing detailed explanations.
②	The article did not include an overview diagram of the overall model structure, which made it difficult for readers to comprehend the architecture of the model.
③	The experiment evaluated accuracy-related metrics such as MAE and MRE but did not provide information on the computation time and performance on large datasets. As a result, readers cannot assess the efficiency of this model.
④	The experimental training methods have the potential to enhance the robustness of the network, but in the end, no robustness test results for the model were provided.
⑤	The experiment did not conduct tests of this framework with different data types of other domains, which makes it difficult to assess the comprehensiveness of its application in various fields.

**Questions:**

①	Why is GgNet only applicable to transductive learning settings in its current form? What are the limitations? How to extend GgNet to support inductive learning?
②	How does the efficiency of Ggnet compare to other models? What is the impact of missing data in the dataset used in the experiment on the experimental results? Can you offer a more detailed explanation of the results?

---

> ### Author Response · Authors · 2023-11-17
>
> We thank the reviewer for the extensive review and positive feedback.
> As for the raised concerns and questions, individual points are addressed below.
>
> **W1/Q3. Discussion of the experimental results**\
> In Sec. 5 of our paper, several paragraphs are dedicated to the discussion of the experimental results.
> Our discussion focuses on the limitations of the existing methods compared to GgNet, which we investigate across different temporal and spatial problem settings.
> We believe that discussing limitations is the most important aspect when interpreting experimental results, as it directly relates to the observed differences in performance.
> From a qualitative angle, Fig. 5 shows results w.r.t. the quality of the prediction and the structure of the learned embeddings.
> However, we agree that further comments might be needed, so we have extended our discussion in Sec. 5.1 of the revised paper. Furthermore, we have performed $4$ additional analyses (see highlighted content in the revised paper Appendix) that significantly contribute to the discussion:
> - in Appendix E.3, we now deepen the disadvantages of graph-based approaches in contrast to GgNet by means of an illustrative example in a geographically remote location;
> - in Appendix E.4, we now show and discuss the GgNet's learned dependencies between variables (intra-location graph $g$);
> - in Appendix E.5, we now investigate location-wise performance and show that the improvement of GgNet with respect to SAITS is spread across locations rather than concentrated around a few locations;
> - in Appendix E.6, we now qualitatively compare uncertainty estimates of GgNet across different scenarios.
>
> Any comment on these revisions would be appreciated.
>
> **W2. Overview diagram of the overall model structure**\
> Please, refer to Fig. 3, which provides an overview of the model's architecture and highlights the different components.
> In the revised paper, we bring modifications to the figure's caption.
> We would appreciate further feedback if, in the reviewer's opinion, any aspect of the figure could be improved.
> Further details about the architecture are provided in Appendix B.
>
>
> **W3. Computation time and performance on large datasets.**\
> Appendix E.1 reports the total computation time of GgNet and relevant baselines, on the largest adopted dataset, i.e., the daily climatic dataset.
> In the revised paper, we extend the pertinent discussion.
> The temporal modeling in GgNet is based on temporal convolutions, which makes it faster than the popular graph-based representative GRIN, which uses a recurrent model instead.
> As for BRITS and SAITS, these are faster than GgNet as they do not account for spatial dependencies.
> The main bottleneck in GgNet is due to the inter-locations graph which makes message-passing operations scale quadratically w.r.t. the number of locations. If scalability to large $N$ is a concern, existing sparse graph learning methods can be considered, e.g., [1].
> However, note that scenarios that involve significantly more sensors than the selected datasets would not likely be sparse and a different set of techniques should be used.
> We thank the reviewer for highlighting this and we have integrated the above considerations into Appendix E.1 of the revision.
>
>
>
>
> **W4/Q2. Robustness test results.**\
> As mentioned in Sec. 4.4, in the first place, we mask out a fraction of the available training channels (entirely for all timestamps in a batch) and train the models to reconstruct such masked portions of data.
> This is a core aspect, as it pushes the training toward learning the virtual-sensing task. Secondly, we also mask out randomly a portion of the input to provide additional supervision to the training routine.
> We recognize that the paper would benefit from a sensitivity analysis of these aspects. As such, we have carried out additional experiments by varying the number of channels being masked out at training time.
> Results are reported in Appendix E.7 of the revised paper.
>
>
> **W5. Different data types of other domains.**\
> The focus of the paper is addressing virtual sensing for sensor networks with poor spatial coverage by proposing a graph-based reconstruction framework.
> For this purpose, we select $3$ datasets from very relevant application domains, which we believe are sufficient to validate the relevance of the proposed methodology and the potential effectiveness of GgNet.
> The application to more general domains and the design of appropriate benchmarks are extremely relevant, yet they go beyond this paper's scope and will constitute interesting future research.

---

> > ### Author Response · Authors · 2023-11-17
> >
> > **Q1. Why is GgNet only applicable to transductive learning settings in its current form?**\
> > GgNet is intrinsically transductive as the node embeddings, which are key to building the inter-location graph and adding local components to different layers, need to be learned for every location from the available data.
> > Despite that, for these types of models, the fine-tuning of new node embeddings can be done very efficiently, as shown in [2].
> > Different strategies can be considered to make GgNet inductive, e.g., by relying on training-free approaches to obtain node embeddings. We thank the reviewer for this question and clarify this in the revision.
> >
> > We hope to have clarified all of the reviewer’s concerns and are happy to provide further details if needed.
> >
> > **References:**\
> > [1] Niculae et al., Discrete latent structure in neural networks. arXiv preprint arXiv:2301.07473, 2023.\
> > [2] Cini et al., Taming Local Effects in Graph-based Spatiotemporal Forecasting, arXiv preprint arXiv:2302.04071, 2023.

---

> > ### Comment · Reviewer_L9oj · 2023-11-18
> >
> > Thanks for the reply. I think the reply addressed my most concerns.

---

### Official Review · Reviewer_CZHK · 2023-11-17

**Soundness:** 2 fair
**Presentation:** 2 fair
**Contribution:** 2 fair
**Rating:** 3
**Confidence:** 5

**Summary:**

This paper proposes a solution to the multivariate spatial interpolation problem through a nested graph representation and graph convolutional networks. Real climate data is used to validate the performance of the proposed solution.

**Strengths:**

1. The problem of spatial interpolation through deep learning is generally meaning for and important.
2. The authors acknowledge traditional spatial interpolation methods such as Kriging
3. Real-world dataset used for experiments.

**Weaknesses:**

1. The claim that Kriging or similar approach won't work due to sparse sensors lacks sufficient justification. The proposed solution can be solved by multivariate Kriging. The authors should at least show the results on such a well-known spatial solution and compare it with their solution.

2. The graph of sensors is built by measuring similarities between node embeddings. However, it is unclear how to learn such embeddings and why it is not affected by sensor sparsity. If two sensors are too far away their values have no correlation. The edge defined in under such a case would be meaningless.

3. There is no demonstration of a successful prediction of missing values on a spatial map.

**Questions:**

1. Why Kriging would not work in this case is not adequately justified. The semi-variogram function can be selected from a variety of options depending on the assumptions. As long as the two locations are not farther than a threshold (the range) apart, they can be assumed to have correlations.

2. Another traditional model that might solve this problem is Markov random field. Why not considering it in the baseline?
/?？
3. How to define the graph edges based on (static) embedding? What information do you used to learn it?

**Details Of Ethics Concerns:**

no concerns.

---

> ### Author Response · Authors · 2023-11-20
>
> We thank the reviewer for the feedback.
>
> As a general comment, we would like to point out that the focus of the paper *is not* on spatial interpolation, but on virtual sensing. The critical difference is that, at a target location, we do not seek to reconstruct a single value but rather one (or more) entire time series.
>
> Furthermore, note that with the term 'spatial dependencies' we do not necessarily refer to physical proximity, but more broadly to generic functional dependencies (learned end-to-end from data, as we discuss below). In the context of our research, 'location' (denoted as $n$) could represent various entities such as a patient from medical data or a material from performance data of different chemistries. As a result, our framework is applicable to generic time series collections, and it is not limited to pre-defined relationships related to physical proximity as in generic spatial modeling.
>
> As for the raised concerns and questions, individual points are addressed below.
>
> **W1/Q1 (1). Limitations of Kriging in sparse settings**\
> Kriging, in general, relies on geographical coordinates, which are used to model spatial dependencies (physical proximity).
> With a dense sensor coverage and high, local, spatial correlation, general kriging methods will perform competitively w.r.t. deep learning methods [1]. However, without the possibility of relying on such pre-defined spatial dependencies, kriging methods might fail. Our method, instead, explicitly learns (end-to-end from data) latent non-linear dependencies among the observed variables both intra and inter-location. This property becomes more important when poor sensor coverage prevents from performing geographical interpolation.
> Lastly, note that OKriging is unsuitable for applications on irregular spatial domains [1], e.g., transportation networks.
>
>
> **W1/Q1 (2). Kriging baseline**\
> While we recognize that adding more standard Kriging baselines would improve completeness, we do argue that it is not a critical element to support our claims. In our study, we benchmark against GRIN, a representative of spatio-temporal Graph Neural Networks (ST-GNN) and a state-of-the-art model for coordinate-based reconstruction. The established literature consistently shows the superior performance of these modern virtual sensing methodologies over standard Kriging in similar settings, e.g., [1,2,3,4].
>
> In particular, note that:
> - Most of the standard kriging methods cannot handle temporal data (fundamental in our setting) while the selected (state-of-the-art) baselines explicitly target these scenarios;
> - Learning a proper spatiotemporal variogram is very computationally demanding, especially for long and multivariate time series;
> - Kriging consists of a *linear interpolation* of the observed data and has a strong Gaussian assumption. More recent models (ours included), instead, are designed to capture *non-linear* dependencies between locations in a learned latent space.
>
> Given the above, we argue that our choice of baselines is coherent with the problem settings and the current state of the art.
>
>
> **W2/Q3. Embedding learning**\
> There might be a critical misunderstanding here. As stated in Sec. 4.2, node embeddings are a table of learnable parameters [5,6] and, as such, are end-to-end learned jointly with the model weights given the task at hand. In our case, node embeddings are learned together with the reconstruction model by minimizing the reconstruction loss. Note that the computations carried out to extract the inter-location graph from such embeddings (Eq. 3) are seamlessly integrated within the end-to-end learning architectures. In other words, the graph is learned end-to-end as well.
>
> Once learned, the embeddings identify each location in a common latent space, of which we show an example in Fig. 5 (right). This placement is not directly influenced by the geographical coordinates of each sensor but is learned exclusively from the observed data (targets and covariates).
> Similarity in such space, then, is not limited to model physical proximity but can capture non-linear dependencies beyond that.

---

> > ### Author Response · Authors · 2023-11-20
> >
> > **W3. Visualization on spatial map**\
> > Generating such visualizations poses challenges in our context. The extensive area that is covered by sparse sensors makes creating spatial maps computationally very demanding. Additionally, our data is multivariate, adding complexity to visualizing all dimensions effectively in a single spatial map. Moreover, as already mentioned, dependencies in our framework extend beyond physical proximity.
> >
> > Nevertheless, we argue that both the main and the appendix provide an in-depth analysis of the empirical results highlighting several aspects of the proposed method such as metrics for each channel and location, robustness, uncertainty estimation, and visualizations of the learned embeddings.
> >
> >
> > **Q2. Why not consider Markov random field**\
> > The extension of Markov random fields to our problem settings is not trivial, as discussed above with respect to standard kriging methods.
> > While we have discussed why we believe that our choice of baselines is sufficient for the purpose of this paper, if the reviewer has a particular method in mind, that would be suitable to our settings, we will be happy to consider including it in the discussion.
> >
> >
> > **References**\
> > [1] Wu, Y. et al., Inductive Graph Neural Networks for Spatiotemporal Kriging, AAAI, 2021.
> >
> > [2] Wu, Y. et al., Spatial aggregation and temporal convolution networks for real-time kriging. arXiv preprint arXiv:2109.12144, 2021.
> >
> > [3] Zheng, C. et al., INCREASE: Inductive Graph Representation Learning for Spatio-Temporal Kriging, Proceedings of the ACM Web Conference, 2023.
> >
> > [4] Appleby, G. et al., Kriging Convolutional Networks, AAAI, 2020.
> >
> > [5] Cini et al., Taming Local Effects in Graph-based Spatiotemporal Forecasting, arXiv preprint arXiv:2302.04071, 2023.
> >
> > [6] Dwivedi, et al., Graph neural networks with learnable structural and positional representations. arXiv preprint arXiv:2110.07875, 2021.

---

### Author Response · Authors · 2023-11-17

We thank again the reviewers for their constructive comments and overall positive evaluation of our paper.
Taking reviewers' feedback into account, we made the following significant additions to the paper:
- in Sec. 5.1, we now present a more thorough discussion of the experimental results;
- in Appendix E, we now present $5$ additional analyses, i.e., a deepening of the disadvantages of graph-based methods for isolated locations, visualization of GgNet's learned intra-location graph, location-wise analysis, visualization of uncertainty estimates, and GgNet's sensitivity to the fraction of masked training channels;
- in Appendix B; we now more extensively discuss scalability and computational time;
- we modified the text to improve clarity of the presentation, e.g., for each contribution stated in the introduction, we add a reference to the corresponding section that supports such a claim.

Revised content is highlighted in orange in the new version of the paper.
We then provide point-by-point answers to each reviewer addressing their concerns.
We look forward to the reviewers' individual responses to our rebuttal.

---

### Meta-Review · Area_Chair_ZuDp · 2023-12-12

**Metareview:**

The paper tackles virtual sensing, namely, the inference of signals at unmonitored locations by exploiting available spatio-temporal measurements coming from existing physical sensors. The authors propose a graph deep learning framework operating on a nested graph structure, which is used to learn dependencies between variables as well as locations.

The biggest concerns remains the lack of comparison to multivariate kriging. Some experimental comparison would help, even if GRIN is indeed a stronger baseline, though the authors did not have enough time to do this during the discussion period. At the very least, a description of to why the framework applies even when interpolation is not possible, or why markov random fields are inapplicable, would strengthen the paper. Finally, though reviewers agree assumptions are not as strong as previously thought, the paper could be revised to better highlight this/avoid confusion.

**Justification For Why Not Higher Score:**

Reviewers maintained concerns.

**Justification For Why Not Lower Score:**

This should be a reject on scores alone, and that's fine if there's no room; that said, scores are not as representative, as a weak reject reviewer did not revise their score, even if they said most of their concerns were addressed, and a very negative reviewer did not engage with the authors at all. I am therefore leaning more positive towards the paper.

---

### Decision · Program_Chairs · 2024-01-16

Accept (poster)